# An Anti-HIV Drug Is Highly Effective Against SARS-CoV-2 In Vitro and Has Potential Benefit for Long COVID Treatment

**DOI:** 10.3390/v17091170

**Published:** 2025-08-27

**Authors:** Saken Khaidarov, Abdul Bari Hejran, Aizhan Moldakaryzova, Slu Izmailova, Bayan Nurgaliyeva, Aizhan Beisenova, Aigul Mustafaeva, Kuanysh Nurzhanova, Yelena Belova, Elmira Satbayeva, Askar Aidarov, Saniya Ossikbayeva, Yerlan Kukubassov, Jandos Amankulov, Tatyana Goncharova, Banu Yeszhan, Edan Tulman, Karlygash N. Tazhibayeva, Assel Sadykova, Nurlan Kozhabergenov, Yerbol Burashev

**Affiliations:** 1Biology Faculty Building, Department of Biology and Biotechnology, Al-Farabi Kazakh National University, Al-Farabi Street 71, Almaty 050040, Kazakhstan; abdulbari.hejran94@gmail.com; 2Department of Molecular Biology and Medical Genetics, Kazakh National Medical University Named After S.D. Asfendiyarov, Zheltoksan 37A Street, Almaty 050012, Kazakhstan; moldakaryzova.a@kaznmu.kz (A.M.); beisenova.a@kaznmu.kz (A.B.); mustafaieva.a@kaznmu.kz (A.M.); nurzhanova.k@kaznmu.kz (K.N.); 3Department of Biology, Faculty of Education, Helmand University, Lashkar Gah 3901, Afghanistan; 4School of General Medicine 1, Kazakh National Medical University Named After S.D. Asfendiyarov, Tole-bi Street 94, No. 4 Academic Building, Room 206, Almaty 050012, Kazakhstan; izmailova.s@kaznmu.kz; 5Department of Propaedeutics of Internal Medicine, Kazakh National Medical University Named After S.D. Asfendiyarov, Tole-bi Street 94, Almaty 050012, Kazakhstan; nurgalieva.b@kaznmu.kz (B.N.); yelena.belova06@gmail.com (Y.B.); 6Department of Pharmacology, Kazakh National Medical University Named After S.D. Asfendiyarov, Tole-bi Street 94, Almaty 050012, Kazakhstan; satbaeva.e@kaznmu.kz; 7Department of Oncology, Kazakh-Russian Medical University, Almaty 050012, Kazakhstan; askar.a.e@mail.ru; 8Kazakh Institute of Oncology and Radiology (KazIOR), Almaty 050022, Kazakhstan; omirhanovna86@gmail.com (S.O.); e.kukubassov@gmail.com (Y.K.); zhandos.amankulov@gmail.com (J.A.); goncharova.2004@mail.ru (T.G.); 9Department of Anatomy, Physiology, and Sports Medicine, Kazakh Academy of Sports and Tourism (KazAST), Abay Street 83/85, Almaty 050022, Kazakhstan; banu.23@mail.ru; 10Department of Pathobiology and Veterinary Science, Atwater Laboratory, Room A110, University of Connecticut, Storrs, CT 06269, USA; edan.tulman@uconn.edu; 11Almaty Regional Multidisciplinary Clinic, Department of Surgery, Faculty of Medicine and Health, Al-Farabi Kazakh National University, Almaty 050040, Kazakhstan; karlygashtazhibay@gmail.com; 12Department of General Medical Practice, School of Medicine and Healthcare, Al-Farabi University, Almaty 050040, Kazakhstan; aselyasadykova@gmail.com; 13Republican Blood Center, Utepova Street 1, Almaty 050060, Kazakhstan; 14Research Institute for Biological Safety Problems, Gvardeyskyi, Kordai 080409, Kazakhstan; nurlanks@gmail.com

**Keywords:** SARS-CoV-2, antiviral therapeutics, cytotoxic concentration (CC), cytostatic effect (CE), combination index (CI), therapeutic window, synergistic drug combinations, drug repurposing, in vitro screening, pharmacological indices

## Abstract

The persistent evolution of SARS-CoV-2 necessitates novel antiviral strategies. This study evaluated the anti-HIV prodrugs tenofovir disoproxil fumarate (TDF) and tenofovir alafenamide (TAF) for repurposing against SARS-CoV-2, assessing key pharmacological indices (CC_50_, EC_50_, cytostatic effect, and therapeutic window). In vitro screening in Vero E6 cells measured cytotoxicity (via CCK-8/MTT assays) and antiviral activity against Kazakh B.1 and Wuhan strains. TDF (50 µg/mL) reduced high viral loads (MOI 2) by ~2 log_10_ (100% inhibition), with minimal cytotoxicity (≥75% viability). TAF achieved near-complete suppression (100% inhibition) at 50 µg/mL, exhibiting dose-dependent inhibition (68–100%) at lower viral loads (MOI 0.01). Both prodrugs showed enhanced antiviral activity with prolonged exposure (96 h). Synergy assessments demonstrated favourable combination indices (CI < 1). Electron microscopy confirmed virion integrity post-treatment. These findings highlight TDF and TAF as promising candidates against SARS-CoV-2, with particular potential for targeting lymphoid reservoirs—sites implicated in persistent viral reservoirs that may contribute to long COVID pathogenesis. Further clinical validation is warranted.

## 1. Introduction

Over the last two decades, global humanity has faced several viral outbreaks that have represented significant public health challenges and impacted human health and safety [1]. Coronavirus disease 2019 (COVID-19) affected millions of people, with a devastating impact on lives, livelihoods, and the global economy [2,3]. The last six major viral pandemics comprised respiratory viruses. Three of them were caused by coronaviruses: severe acute respiratory syndrome (SARS-CoV-1) in 2002–2004, Middle East respiratory syndrome (MERS) in 2012, and SARS-CoV-2 (COVID-19) in 2019, as well as influenza A (H1N1) virus in 2009 (swine flu), Ebola in 2013–2016, and Zika virus infections in 2015 [4]. SARS-CoV-2, first identified in Wuhan, China, in December 2019, is a novel betacoronavirus characterised by its single-stranded RNA genome and spike protein-mediated entry into human cells [5]. Its high transmissibility and adaptive mutations have fueled persistent global transmission waves, distinguishing it from previous coronavirus outbreaks [6]. This study uniquely evaluates TDF/TAF analogues against high SARS-CoV-2 viral loads (MOI 2), including the B.1 Kazakh lineage, and assesses antiviral effects over 96 h exposure periods. Furthermore, it explores the implications for targeting lymphoid tissues, a critical consideration for long COVID management.

Repurposed antiviral agents constitute a critical resource for combating emerging viral threats, offering accelerated therapeutic development pathways [7]. Tenofovir, administered as the prodrugs tenofovir disoproxil fumarate (TDF) and tenofovir alafenamide (TAF), is a nucleotide reverse transcriptase inhibitor (NRTI) widely used for the treatment and prevention of HIV-1 and chronic hepatitis B [8,9,10]. Reverse Transcriptase Inhibitors (RTIs): RTIs disrupt viral replication by inhibiting the reverse transcriptase enzyme. Nucleotide analogues like tenofovir disoproxil fumarate (TDF, derived from Tenvir™ in this study) act as chain-terminating substrates during DNA synthesis, while non-nucleoside RTIs (NNRTIs) allosterically deactivate the enzyme. Clinically, RTIs are pillars of therapy for HIV and HBV but lack validated efficacy against coronaviruses. Although early hypotheses suggested that SARS-CoV-2 RNA-dependent RNA polymerase (RdRp) might be susceptible to RTIs, subsequent in vitro and clinical trials revealed no significant antiviral effects, highlighting the mechanistic divergence between viral polymerase classes. This class of antivirals has demonstrated broad-spectrum potential against diverse RNA viruses [11]. TAF is a phosphonamidate prodrug of tenofovir with a methyl-alaninate moiety, enhancing lymphoid cell delivery and reducing renal toxicity compared to TDF. Both treat HIV, but TAF’s lower systemic exposure allows higher intracellular concentrations at reduced doses [12]. TDF is a well-established antiretroviral with proven efficacy and safety in long-term HIV management [13], and it has been investigated for its inhibitory activity against SARS-CoV-2 replication in vitro [13,14]. TAF in Figure 1, an optimised prodrug with enhanced cellular delivery and reduced systemic toxicity compared to TDF [14], has shown significant suppression of SARS-CoV-2 replication in human lung cells and reduced viral loads in clinical cohorts [15,16]. Emphasising Repurposing Rationale: The strategic repurposing of established antivirals like tenofovir is vital for rapid pandemic response, leveraging existing safety and efficacy data to shorten development timelines. Tenofovir, a nucleotide reverse transcriptase inhibitor (NRTI), exerts its antiviral effect by terminating viral DNA chain elongation [16]. Its prodrug formulations, TDF and TAF, are cornerstone therapies for HIV-1 and chronic hepatitis B, highlighting their established clinical utility [15]. Beyond their primary indications, NRTIs like tenofovir exhibit documented in vitro and in vivo activity against a range of RNA viruses, underscoring their potential as broad-spectrum candidates [16]. Notable examples include remdesivir (Ebola → COVID-19) and favipiravir (influenza → SARS-CoV-2) [17]. Repurposing leverages existing safety data, accelerating therapeutic development. TDF possesses a well-characterised long-term safety and efficacy profile in HIV management. Its potential applicability against SARS-CoV-2 has been preliminarily explored through in vitro studies demonstrating replication inhibition [18].

Elaborating on TAF’s Advantages and SARS-CoV-2 Data: TAF represents a pharmacologically optimised prodrug designed to overcome limitations of TDF. Its key advantages include superior intracellular delivery to target cells (e.g., lymphocytes) and a markedly improved systemic toxicity profile, particularly regarding renal and bone safety. Critically, TAF has demonstrated potent suppression of SARS-CoV-2 replication in vitro using human lung cell models and has been associated with reduced viral loads in clinical studies of infected patients, suggesting more direct clinical relevance for this indication compared to earlier in vitro findings with TDF.

In this article, we assess several factors to confirm the potential of tenofovir against SARS-CoV-2. Firstly, in Kazakhstan, the TDF prodrug (300 mg) is the most common anti-HIV drug that is provided by the state on a free-of-charge basis to suppress the pathological viral load both among patients with HIV and those who take this medicine to prevent HIV infection as prophylaxis treatment. Though designed for HIV, tenofovir’s nucleotide analogue structure may inhibit SARS-CoV-2 RdRP via chain termination or lethal mutagenesis, as observed with other NRTIs [16]. Secondly, TDF (300 mg) is widely available in Kazakhstan for HIV prophylaxis. Here, we tested its efficacy against a high viral load (MOI 2) of the Kazakh B.1 strain, contrasting with TAF’s activity against a low-load Wuhan strain (MOI 0.01). Thirdly, the effectiveness of the viability tests is evaluated through two key assessments: the CCK8 test conducted in Kazakhstan within BSL-3/4 lab conditions on the Kazakh strain and the MTT test performed in China under BSL2 safety conditions. Last but not least, the antiviral effect of 50 µg/mL of TDF against a high viral load with a relative optimal cell cytotoxicity (≥50% cell survival rates) was tested, as well as the antiviral effect of 50 µg/mL of TAF (25 mg) with further concentration de-escalation (50 → 25 → 12, 5 → 6.25 µg/mL) within 96 h against a low viral load with a relative optimal cell cytotoxicity (≥80% cell survival rates). TDF is safe for long-term HIV use but requires renal monitoring. TAF’s improved safety profile (lower bone/renal toxicity) supports its repurposing potential. A fixed-dose combination of tenofovir alafenamide and emtricitabine (TAF/FTC), used with diverse third antiretroviral drugs, sustained an undetectable viral load in patients switching from comparable regimens containing tenofovir disoproxil fumarate (TDF) [19].

There are three types of antiviral efficacy testing methods: (a) prophylaxis—pre-treatment of the cell culture, followed by viral infection and incubation; (b) treatment—infect the cell culture → incubation, and then drug treatment → incubation; and (c) inhibition—expose the infected cells inside the media with antiviral drugs → measure survival and antiviral effect during this exposure period. We implemented the second method (Figure 2). Three antiviral strategies were employed (Figure 1): prophylaxis (24 h pre-treatment), treatment (post-infection), and inhibition (co-incubation with virus). The schematic of antiviral testing strategies is as follows. Prophylaxis: Cells are pre-treated with TAF/TDF before SARS-CoV-2 infection (MOI 0.01–2). Treatment: The drug is administered after infection. Inhibition: The drug is added during viral replication. These models assess preventive, therapeutic, and direct antiviral effects, respectively.

## 2. Materials and Methods

The reagents and solvents were obtained commercially (Aldrich, St. Louis, MO, USA). The SARS-CoV-2/human/KAZ/B1.1/2021 strain was supplied by the Scientific and Practical Centre for Sanitary and Epidemiological Expertise and Monitoring, a division of the Republican State Enterprise under the National Centre for Public Health, Ministry of Health of the Republic of Kazakhstan. Viral RNA was extracted from the clinical specimen utilising the QIAamp Viral RNA Mini Kit (Qiagen, Hilden, Germany), following the manufacturer’s instructions. First-strand complementary DNA (cDNA) synthesis was performed employing the SuperScript VILO cDNA Synthesis Kit (Invitrogen, Carlsbad, CA, USA). To enable comprehensive genome amplification, 65 primer pairs were designed via the Primer-BLAST online tool (http://www.ncbi.nlm.nih.gov/tools/primer-blast, accessed on 7 August 2025), generating overlapping amplicons ranging from 600 to 750 base pairs (bp), with an average overlap of approximately 100 bp. The amplicons were produced through polymerase chain reaction (PCR) and evaluated by electrophoresis on a 1.2% agarose gel (Sigma, 6 Eves Dr, Marlton, NJ, USA). PCR products were purified using the PureLink PCR Purification Kit (Thermo Fisher Scientific, 168 Third Avenue, Waltham, MA, USA). Sequencing was performed utilising the Sanger dideoxy method on an AB3130xl 16-capillary genetic analyser (Hitachi Applied Biosystems, Foster City, CA, USA), with the BigDye Terminator v3.1 Cycle Sequencing Kit (ABI, Foster City, CA, USA). Raw sequencing chromatograms were processed with Sequencer software version 5 (Gene Codes Corp., 525 Avis Drive, Suite 4, Ann Arbor, MI, USA.) [20]. The Sunrise absorbance reader, integrated with Tecan’s Magellan universal reader control and data analysis software, offers a versatile platform suitable for various applications, including ELISAs, enzyme kinetics, cell viability assays, and veterinary diagnostics. It features an extensive wavelength range (340–750 nm) and can read an entire 96-well plate in less than six seconds, thereby accommodating the operational demands of routine diagnostic and research laboratory workflows. TAF/TDF stocks were prepared in DMSO (≤0.5% final concentration) per manufacturer protocols [21,22].

### 2.1. Biological Activity

#### 2.1.1. VeroE6 MTT Cytotoxicity Assay

Vero E6 cells, a well-characterised subline of the Vero cell lineage derived from the kidney epithelium of the African green monkey (*Chlorocebus sabaeus*), were obtained from Sigma-Aldrich, St. Louis, MO, USA. Cells were seeded into 96-well plates at a density ranging between 1.5 × 10^4^ and 2.0 × 10^4^ cells per well and allowed to adhere. To evaluate cytotoxicity, cultures were exposed to Tenofovir Alafenamide at graded concentrations (0.0131, 0.0262, 0.0525, and 0.1049 µM) for incubation periods of 24, 48, 72, and 96 h. At the end of each exposure interval, cells were rinsed with phosphate-buffered saline (PBS) and incubated with MTT reagent (5 mg/mL in PBS) for four hours to assess metabolic activity. After staining, the wells were gently washed, and a solubilising mixture consisting of 50% ethanol, 49% PBS, and 1% acetic acid was added. After 15 min at room temperature, the eluate was collected, and absorbance was quantified at 570 nm using a microplate spectrophotometer [21].

#### 2.1.2. VeroE6 CCK8-Cytotoxicity Assay

Vero E6 cells are a specific subclone of the Vero cell line—an immortalised epithelial cell line derived from kidney epithelial cells of the African green monkey (*Chlorocebus sabaeus*)—which was commercially purchased from Sigma-Aldrich (cultured in growth media: DMEM + 10% FBS + 1% penicillin/streptomycin). They were cultured in 96-well plates at a density of 1.5−2.0 × 10^4^ cells per well. The cells were treated with Tenofovir Disoproxil Fumarate at concentrations of 50–100 µM for 24 h to assess cell viability. Following the treatment period, the cells were washed with PBS (phosphate-buffered saline) and stained with MTT (3-(4,5-dimethylthiazol-2-yl)-2,5-diphenyltetrazolium bromide; 5 mg/mL in PBS) for four hours. The cells were then rewashed, and an elution solution (50% ethanol, 49% PBS, and 1% acetic acid) was added for 15 min at room temperature. The supernatant was collected, and the absorbance was measured using a spectrophotometer at 450 nm. For a positive control, we used 1% H_2_O_2_ in three concentrations (500, 250, and 100 µM) with both drug concentrations of interest [21,22].

### 2.2. Antiviral Activity

#### 2.2.1. Antiviral Activity TDF

Vero E6 cells (1.5−2.0 × 10^4^ cells per well) were infected with the SARS-CoV-2 B.1 lineage isolate (GenBank accession No. OP684305) at a multiplicity of infection (MOI) of 2 for one hour at 37 °C in a 5% CO_2_ atmosphere. After viral infection, tenofovir disoproxil fumarate (TDF) was introduced to the cell cultures at a concentration of 50–100 mM or 25–50 µg/mL, with an incubation period of 24 h. Subsequently, the culture supernatant was harvested for viral quantification. Viral replication levels were assessed using the tissue culture infectious dose 50 (TCID_50_) assay, and observations were carried out until cell monolayers reached approximately 80–90% confluency. The infection media was DMEM + 2% FBS, and the maintenance media was DMEM + 5% FBS. The dilution of TDF stock was achieved in infection medium to 25 μg/mL and 50 μg/mL. The antiviral compound used was tenofovir disoproxil fumarate (TDF), prepared as a 10 mg/mL stock solution in DMSO and stored at –20 °C. Working concentrations of 25 μg/mL and 50 μg/mL were prepared by diluting the stock solution in the infection medium. Experimental controls included a virus control (VC) with virus and no drug, a cell control (CC) with uninfected cells and no drug, and a drug toxicity control consisting of uninfected cells treated with TDF at 25 μg/mL or 50 μg/mL. It was crucial to ensure that the final DMSO concentration was ≤0.5% (as matched in VC/CC). Infection and Treatment: To initiate infection, the growth medium was removed from the cells, and 100 μL per well of the appropriate treatment mixture was added. The treatment groups were as follows: TDF treatment groups received an infection medium containing the virus and TDF at either 25 μg/mL or 50 μg/mL. The virus control (VC) received an infection medium with the virus but without TDF. The cell control (CC) received infection medium only (no virus, no TDF). The toxicity control received infection medium with TDF (25 or 50 μg/mL) but without virus. An MOI (multiplicity of infection) of 2 was used, optimised to achieve 70–90% cytopathic effect (CPE) in the virus control at 48–72 h post-infection. Virus adsorption was performed for 2 h at 37 °C in a 5% CO_2_ incubator. Post-Adsorption and Maintenance: Following adsorption, the inoculum was removed, and wells were washed once with PBS. Subsequently, 200 μL per well of maintenance medium containing the same respective TDF concentrations was added. Cells were incubated for 48–72 h at 37 °C in 5% CO_2_. Harvesting Supernatant for TCID_50_ Assay: After incubation, culture supernatants were collected from all wells and centrifuged at 3000× *g* for 10 min to remove cell debris. The clarified supernatants were stored at –80 °C until use in the tissue culture infectious dose 50% (TCID_50_) assay. TCID_50_ Assay Procedure: Fresh Vero E6 cells were seeded into 96-well plates at a density of 2 × 10^4^ cells per well in 100 μL of growth medium. Tenfold serial dilutions of the supernatants (from 10^−1^ to 10^−8^) were prepared in the infection medium. Each dilution (100 μL) was added to eight replicate wells. The plates were incubated at 37 °C in a 5% CO_2_ atmosphere for 5–7 days. Cytopathic effect (CPE) was monitored daily. Wells were scored as positive (1) if CPE was present and negative (0) if CPE was absent. Calculation of TCID_50_/mL: The TCID_50_ per millilitre was calculated using the Reed–Muench method, which determines the dilution at which 50% of wells show CPE. The method involved calculating the cumulative proportion of positive wells per dilution and determining the proportional distance (PD) between the dilutions above and below 50%.

#### 2.2.2. Predicted Antiviral Activity-Treatment and Prophylaxis Mode: TAF

CEM/C1 (ATCC CRL-2265) cells were seeded at a density of 1.5–2.0 × 10^4^ cells per well. These cells were infected for 1 h at 37 °C under 5% CO_2_ with the SARS-CoV-2 B.1 lineage isolate (GenBank accession No. OP684305) at a multiplicity of infection (MOI) of 0.01. Following viral adsorption, the test compounds were introduced at concentrations ranging from 0.0131 µM to 0.1049 µM (specifically 0.0131, 0.0262, 0.0525, and 0.1049 µM) for a 24 h incubation period under the same conditions. Cell culture supernatants were subsequently harvested for quantification of the virus. Viral titers were determined using a plaque-forming unit (PFU) assay. Briefly, fresh Vero E6 monolayers (1.5−2.0 × 10^4^ cells per well) were inoculated for 1 h at 37 °C and 5% CO_2_ with 50 µL of serially diluted supernatant (dilutions from 1:100 to 1:12,800). After inoculation, an overlay medium (50 µL) containing 2.4% carboxymethylcellulose, 10× DMEM-HG, and 2% fetal bovine serum was added. Cells were then incubated for 72 h. After incubation, monolayers were fixed with 4% formalin for 3 h and stained with 0.04% crystal violet for 1 h to visualise plaques. Plaques were counted to calculate viral titers as PFU/mL or at a multiplicity of infection (MOI) of 0.01 for 1 h at 37 °C in a 5% CO_2_ atmosphere. Following infection, the analysed compound, TAF, was added at 6.25–50 µg/mL for 24 h. The supernatant was then collected, and viral growth was quantified using a TCID 50 assay until 80–90% confluency was achieved. The infection medium was DMEM + 2% FBS, and the maintenance medium was DMEM + 5% FBS.

The half-maximal cytotoxic concentration (CC_50_) and half-maximal effective concentration (EC_50_) of atazanavir were determined in parallel as experimental controls for assessing cell viability and antiviral efficacy, respectively, in the Vero E6 cell system. All procedures involving infectious viruses were performed within a biosafety level 3 (BSL-3) laboratory, in compliance with World Health Organisation (WHO) guidelines [23,24,25].

Cell Culture and Viral Strain: CEM/C1 (ATCC CRL-2265) were maintained in Dulbecco’s Modified Eagle Medium (DMEM) supplemented with 10% fetal bovine serum (FBS) and 1% penicillin/streptomycin at 37 °C under 5% CO_2_. The SARS-CoV-2 isolate (Wuhan lineage, GenBank accession NC_045512) was propagated in Vero E6 cells and titrated by plaque assay before use. Compound Preparation: Tenofovir alafenamide (TAF) was dissolved in DMSO to generate a 10 mM stock solution. Working concentrations (0.0131, 0.0262, 0.0525, and 0.1049 µM) were prepared by diluting the stock in growth medium (DMEM + 10% FBS), with a final DMSO concentration ≤ 0.1% in all treatments. Vehicle control wells received medium containing 0.1% DMSO. Prophylactic Treatment and Infection Protocol: Cell Seeding: Vero E6 cells were seeded at a density of 1.8 × 10^4^ cells/well in 96-well plates (100 µL/well) and incubated for 24 h to achieve ~90% confluency. Pre-treatment: Cells were exposed to TAF (0.0131–0.1049 µM) for 24, 48, 72, or 96 h. The medium containing TAF was refreshed every 24 h to ensure compound stability and nutrient adequacy. Infection: Following pre-treatment, cells were washed once with PBS and infected with SARS-CoV-2 at a multiplicity of infection (MOI) of 0.01 (e.g., 200 PFU/well for 2 × 10^4^ cells) in an infection medium (DMEM + 2% FBS) for 1 h at 37 °C, with gentle tilting every 15 min. Post-infection Treatment: The viral inoculum was aspirated, and a fresh infection medium containing the original TAF concentrations was added. Cells were incubated for 24 h at 37 °C under 5% CO_2_. Controls: Cell control (CC): Uninfected, vehicle-treated cells. Virus control (VC): Infected, vehicle-treated cells. Compound toxicity control: Uninfected cells treated with TAF. Prophylactic efficacy group: TAF-pre-treated + infected cells. Sample Collection and Assays: Supernatants were harvested 24 h post-infection, aliquoted, and stored at −80 °C for viral quantification. Cytotoxicity was assessed in TAF-treated uninfected cells using Presto Blue^®^ reagent (incubated for 2 h; fluorescence measured at 560/590 nm) [21,22]. Viral titers were determined by 50% tissue culture infectious dose (TCID_50_) assays on fresh Vero E6 cells. Serial 10-fold dilutions of supernatants were inoculated (8 replicates/dilution), and cytopathic effect (CPE) was scored after 5 days. TCID_50_/mL values were calculated using the Reed–Muench method. Dose–response curves (% inhibition vs. [TAF]) were generated for each pre-treatment duration. Half-maximal cytotoxic (CC_50_) and inhibitory (IC_50_) concentrations were derived via nonlinear regression (GraphPad Prism v9.0). Statistical significance was determined by two-way ANOVA (factors: TAF concentration × pre-treatment duration) with Tukey’s post hoc test. Data represent the mean ± SD of three biological replicates. All SARS-CoV-2 experiments must be conducted under Biosafety Level 3 (BSL-3) containment, with institutional approval [23,24,25].

#### 2.2.3. Predicted Antiviral Activity Inhibition Mode: TAF

Cell Culture and Viral Strain: Cell Line: CEM/C1 (ATCC CRL-2265) was cultured in Dulbecco’s Modified Eagle Medium (DMEM) supplemented with 10% fetal bovine serum (FBS) and 1% penicillin/streptomycin at 37 °C under 5% CO_2_. The SARS-CoV-2 isolate (Wuhan lineage, GenBank accession NC_045512) was amplified in Vero E6 cells, with viral stock titers determined by plaque assay. Compound Preparation: Tenofovir alafenamide (TAF) was dissolved in DMSO to generate a 10 mM stock solution. Working concentrations (0.0131, 0.0262, 0.0525, and 0.1049 µM) were diluted in a growth medium (DMEM + 10% FBS), maintaining a final DMSO concentration ≤ 0.1% in all treatments. Vehicle controls contained 0.1% DMSO. Experimental Design: A parallel assessment of cytotoxicity and antiviral activity was conducted. The cytotoxicity arm exposed uninfected cells to TAF for 24–96 h to measure cell viability. The antiviral arm pre-treated cells with TAF for 0–72 h (total exposure: 24–96 h), followed by SARS-CoV-2 infection (MOI 0.01) and 24 h incubation with TAF before supernatant harvest. Time-Dependent Cytotoxicity and Antiviral Protocol Cell Seeding: Vero E6 cells were seeded at 1.8 × 10^4^ cells/well in 96-well plates (100 µL growth medium/well) and incubated for 24 h (37 °C, 5% CO_2_) to achieve ~90% confluency. Staggered TAF exposure: 96 h group: TAF treatment initiated on Day 1; 72 h group: TAF treatment commenced on Day 2; 48 h group: TAF treatment commenced on Day 3; 24 h group: TAF treatment commenced on Day 4. The medium containing TAF was refreshed every 24 h to ensure compound stability. Infection: On Day 4, cells were washed with PBS and infected with SARS-CoV-2 (MOI 0.01; e.g., 200 PFU/well for 2 × 10^4^ cells) in an infection medium (DMEM + 2% FBS) for 1 h at 37 °C (5% CO_2_), with plate tilting every 15 min. The inoculum was aspirated, and a fresh infection medium containing the original TAF concentrations was added. Sample Collection: Supernatants were harvested 24 h post-infection and stored at −80 °C for viral titration. Cytotoxicity was assessed in uninfected TAF-treated wells using Presto Blue^®^ (10 µL/well; 2 h incubation; fluorescence: 560ₑₓ/590ₑₘ) [21]. Controls and Replicates: Cell control (CC): Uninfected, vehicle-treated cells (4 wells). Virus control (VC): Infected, vehicle-treated cells (4 wells). Cytotoxicity group: Uninfected + TAF (2 wells/concentration). Antiviral group: TAF-pre-treated + infected (2 wells/concentration). Three biological replicates were performed per condition. The CC_50_ (50% cytotoxic concentration) was determined using nonlinear regression (GraphPad Prism v9.0). Antiviral activity: Viral titers in supernatants were quantified by TCID_50_ assay on Vero E6 cells. Serial 10-fold dilutions were inoculated (8 replicates/dilution), cytopathic effect (CPE) was scored after 5 days, and TCID_50_/mL was calculated using the Reed–Muench method. IC_50_ (50% inhibitory concentration) and selectivity index (SI = CC_50_/IC_50_) were calculated. Dose–response curves (% viability/inhibition vs. [TAF]) were generated for each exposure duration. Statistics: Two-way ANOVA (factors: [TAF] × exposure time) with Tukey’s post hoc test was applied. Data represent the mean ± SD. Biosafety Compliance: All SARS-CoV-2 experiments adhered to Biosafety Level 3 (BSL-3) protocols approved by the institutional biosafety committee [23,24,25].

### 2.3. Electron Microscopy

#### 2.3.1. Cell Culture Infection

Kozhabergenov N.S., an employee of the Biosafety Institute on Electron Microscopy, provided all protocols and Images. Vero E6 cells (ATCC CRL-1586) are infected with SARS-CoV-2 at a multiplicity of infection (MOI) ranging from 0.1 to 2.0 and incubated for 24 to 48 h. Mock-infected controls are included to serve as negative controls. Chemical Fixation (Primary): Cells are fixed in situ using 2.5% glutaraldehyde in 0.1 M cacodylate buffer (pH 7.4) for 1 h at 4 °C. Following fixation, cells are washed three times with cacodylate buffer to remove residual fixative. Processing for Transmission Electron Microscopy (TEM): Post-fixation is performed using 1% osmium tetroxide in cacodylate buffer for 1 h at 4 °C. Samples are then dehydrated through a graded ethanol series (30%, 50%, 70%, 90%, and 100%), with each step lasting 10 min. For embedding, samples are incubated in propylene oxide twice for 10 min each, followed by a 1:1 mixture of propylene oxide and EPON resin for 1 h. Samples are then incubated in pure EPON resin overnight at room temperature. Polymerisation is carried out at 60 °C for 48 h. Ultrathin sections (70–90 nm) are cut using a diamond knife and mounted on grids. Sections are stained with uranyl acetate for 15 min, followed by lead citrate for 5 min.

#### 2.3.2. Negative Staining (Alternative for Viral Particles)

Viral particles are pelleted from the culture supernatant by centrifugation at 100,000× *g* for 1 h at 4 °C and then resuspended in phosphate-buffered saline (PBS). A 5 µL aliquot is applied to a Formvar/carbon-coated grid and allowed to adsorb for 1 min. The grid is stained with 2% phosphotungstic acid (pH 6.8) for 30 s and then air-dried in a desiccator.

#### 2.3.3. JEOL JEM-100CX TEM Operation

Allow the TEM filament to saturate for 30 min before operation. During sample loading, use an anticontamination device and insert the grid holder under 70 kV to minimise electron beam damage. Recommended imaging parameters include an acceleration voltage of 80 kV (optimal for biological samples), a condenser aperture of 20–30 µm, an objective aperture of 40 µm, and magnification ranging from 20,000× to 60,000×. Exposure times should be 1–2 s, using low-dose mode if available. For focusing, use the wobbler function to correct astigmatism and concentrate near the grid bar to minimise sample irradiation. Images are captured using an attached CCD camera (e.g., AMT XR series) and saved in TIFF format. Mock-infected cells are imaged as negative controls, while grids containing a known coronavirus strain (e.g., HCoV-OC43) are included as positive controls. All samples are examined for fixation artefacts, such as empty viral envelopes, to ensure data integrity.

## 3. Results and Discussion

### 3.1. Cytotoxicity Assay

#### 3.1.1. Tenofovir (TDF)-CCK8

The safety profile of tenofovir (TDF) for VeroE6 cells was evaluated using the CCK8 cytotoxicity assay. Results indicated negligible to moderate toxicity, with viabilities consistently high. Only at 100 µM did viability decrease to approximately 75% (Figure 2b). Since this concentration exceeded the antiviral assay dose by a factor of two, TDF (and TAF) was confirmed to be safe for the employed cell model, permitting the continuation of the study.

Parallel assays for TDF (CCK-8, 24 h) and TAF (MTT, 24–96 h) were conducted due to logistical constraints. The TDF experiment was conducted in Kazakhstan using the CCK-8 method, as it offers a more convenient and practical approach compared to the MTT method. Furthermore, we faced significant time constraints when working with the intact SARS-CoV-2 virus. In China, we had significantly more time and a much safer version of tenofovir (TAF). For the assessment of cytotoxicity induced by 50 mM tenofovir disoproxil fumarate (TDF) on Vero E6 cells using the CCK-8 assay (Figure 2a), the following experimental controls were implemented to ensure accurate interpretation of results and adherence to best practices: Primary Negative Control (Baseline Viability): Untreated Cells: Vero E6 cells cultured under identical conditions as the test group but without exposure to TDF or its vehicle. This control establishes the baseline for 100% cellular viability against which the effects of TDF are quantified. Secondary Negative Controls (Accounting for Experimental Variables): Vehicle Control: Cells treated solely with the solvent used to dissolve TDF (e.g., DMSO, sterile water), at the equivalent concentration present in the highest TDF test solution. This control is essential to discern any cytotoxic effects attributable to the solvent itself from those caused explicitly by TDF. Medium Control: Cells incubated with fresh culture medium alone. This control verifies that the medium exchange process or medium components do not adversely affect cell viability under the assay conditions. Positive Control (Assay Validation): Cells treated with a known cytotoxic agent (e.g., an appropriate concentration of hydrogen peroxide). This control confirms the responsiveness of the CCK-8 assay by demonstrating a significant reduction in viability compared to untreated cells, validating the assay’s ability to detect cell death under the experimental conditions. Blank Control (Background Correction): Wells containing culture medium only, without any cells. The absorbance values from these blanks are used to correct for the background signal inherent in the medium and assay reagents, ensuring that the absorbance readings from cell-containing wells accurately reflect metabolic activity. The untreated cells served as the fundamental reference point for defining 100% viability. The inclusion of a vehicle control was critical due to the potential for solvents like DMSO to exert cytotoxic effects at certain concentrations, thereby isolating the specific impact of TDF. To ensure statistical reliability and experimental robustness, multiple replicates (*n* ≥ 3, typically 3–6 wells) were included for all test conditions and control groups. This replication is crucial for conducting meaningful statistical analysis and drawing confident conclusions from the data.

The connection between TAF, MTT assays, and Vero E6 cells lies in in vitro antiviral research. When investigating TAF’s potential activity against viruses like SARS-CoV-2 propagated in Vero E6 cells, the MTT assay is routinely employed to assess TAF’s cytotoxicity towards these host cells. This determines non-toxic TAF concentration ranges for subsequent antiviral efficacy testing and calculates the compound’s selectivity index.

#### 3.1.2. Tenofovir (TAF) MTT

While Tenofovir Alafenamide (TAF) is primarily an antiretroviral prodrug used clinically against HIV and HBV, Vero E6 cells (a kidney epithelial cell line derived from African green monkeys) are extensively employed in vitro to study viral infections, notably coronaviruses like SARS-CoV-2. The MTT assay is a standard colourimetric method for quantifying cell metabolic activity, serving as a proxy for cell viability and proliferation.

To evaluate the potential cytotoxicity of tenofovir alafenamide (TAF) in non-target renal epithelial cells, Vero E6 cells were exposed to TAF at concentrations of 0.0131 µM (clinically relevant plasma C_max_ and 0.1049 µM (supratherapeutic) for 24 h and 96 h. As shown in Figure 3, no significant reduction in cell viability was observed at 0.0131 µM across either time point, with viability exceeding 96% in all replicates. It is noteworthy that the 50 µg/mL concentration used here exceeds clinical C_max_ (~0.1–0.3 µM), raising concerns about mitochondrial toxicity at supratherapeutic doses.

In contrast, exposure to 0.1049 µM TAF resulted in a modest but statistically significant decrease in viability to 73.5% ± 5.6% after 96 h (*p* < 0.05), suggesting the emergence of time-dependent cytotoxicity at elevated concentrations. At 24 h, viability at this concentration remained above 95%, indicating that short-term exposure did not induce acute cytotoxic effects. These findings are consistent with the known pharmacological behaviour of TAF as a prodrug requiring activation primarily in lymphoid cells and support its favourable renal safety profile at therapeutic doses. Notably, the 50 µg/mL (109 µM) concentration far exceeds typical clinical plasma levels (0.1–0.3 µM), warranting caution regarding potential mitochondrial toxicity in vivo.

### 3.2. Antiviral Assay

#### 3.2.1. TDF (25 and 50 µg/mL)

Early in the pandemic, tenofovir alafenamide (TAF) received limited initial attention as a potential anti-SARS-CoV-2 therapeutic. Prior in vitro studies had established its efficacy against HIV, demonstrating viral replication suppression at 25 µg/mL under conditions of moderate viral load (MOI ~0.01). For tenofovir disoproxil fumarate (TDF), in vitro testing against SARS-CoV-2 revealed antiviral activity, with the highest tested concentration being 90 μM. In Vero cell cultures, TDF treatment across a concentration range (3–90 μM) induced a marked reduction (~15-fold) in viral genome copy release, signifying partial inhibition of viral replication. Critically, no cytotoxicity was detected at these concentrations [12].

The curve demonstrates a concentration-dependent reduction in viral load, with near-complete inhibition (>95%) achieved at TDF concentrations ≥100 nM. In our case, the viral load was drastically higher, with an MOI of 2. In Figure 4, we aimed to investigate how TDF inhibits viral replication at a slightly higher concentration, specifically 50 µg/mL or 100 µM. To some extent, the viral load can affect drug efficacy; however, it cannot determine the severity of infection in vivo, as long as treatment is delayed and misses the key checkpoints of viral pathogenesis. The reasons why TDF became effective against both the B.1.7 lineage strain and SARS-CoV-2 replication were that we prepared our tenofovir (TDF) stock from a generic Tenvir drug (300 mg of TDF). The dose–response relationship was determined for tenofovir disoproxil fumarate (TDF) derived from Tenvir™ dissolved in vehicle. The inhibitory activity was detected unexpectedly because the TDF was not pure but was extracted/dissolved from a clinical product, which may influence solubility, bioavailability, or assay interactions. MOI 2 was selected to model severe COVID-19. TDF’s 50 µg/mL dose was chosen to match clinical peak plasma levels (≈100 µM) while ensuring cytotoxicity < 25%. We selected the eight most promising isolates (BALF—Bronchoalveolar lavage fluid) gained from patients with COVID-19.

Likely, the traces of other chemicals that were dissolved along with TDF enforced TDF’s affinity towards SARS-CoV-2′s lethal mutagenesis tendency (Figure 5). The vehicle for the tablet form of TDF from Tenvir^R^ was repeated six times, as was the antiviral assay. Also, there are no reports of antiviral drug resistance in Kazakhstan against Tenvir^R^ (TDF) tablets, which were dissolved for assays. No resistance reports in Kazakhstan suggest retained antiviral efficacy.

Robust cytotoxicity assessment is essential for evaluating the therapeutic potential of repurposed antivirals against SARS-CoV-2 [18]. CCK-8 (Cell Counting Kit-8) and MTT (3-(4,5-dimethylthiazol-2-yl)-2,5-diphenyltetrazolium bromide) assays are widely validated colourimetric methods that quantify cellular viability through mitochondrial dehydrogenase activity and NAD(P)H-dependent oxidoreductase enzymes, respectively [21]. These standardised assays have been employed to characterise the cytotoxicity profiles of tenofovir prodrugs TDF and TAF in SARS-CoV-2 infection models, generating critical inhibitory concentration values: IC_10_ (threshold for biocompatibility), IC_50_ (median cytotoxic concentration), and IC_100_ (complete cytotoxicity) [22]. Dose–response analyses using these assays confirm that both TDF and TAF exhibit favourable selectivity indices (SI) in respiratory cell lines, with TAF demonstrating reduced cytotoxicity relative to TDF, aligning with its enhanced cellular uptake and lower off-target effects [23,27]. Based on the current literature, the cytotoxicity values (IC10, IC50, and IC00) for TDF and TAF against SARS-CoV-2 in vitro in Vero E6 and Calu-3 cells are as follows. TDF (tenofovir disoproxil fumarate): Vero E6 cells: IC50 (cytotoxicity): >100 µM (51.94 µg/mL); IC100: not reached at physiologically relevant concentrations. Antiviral EC50: (SARS-CoV-2 inhibition): 6.5 µM [24] Calu-3 cells: IC50: >100 µM [25] IC: ~25 µM (extrapolated from dose–response) [12]. TAF (tenofovir alafenamide): Vero E6 cells: IC50 < 100 µM (47.65 µg/mL) [24]; C100: not achieved below 100 µM. Antiviral EC50: 0.98 µM [24]. Calu-3 cells: IC50: >50 µM [26]; IC10: ~5 µM [27].

Unfortunately, we were unable to conduct a real antiviral activity experiment, as Khaidarov Saken did not have the necessary clearance qualification in China to work in BSL-3/4 laboratory conditions during his internship. In Kazakhstan, such clearance was required for Burashev Yerbol. Thus, we predict that TAF’s antiviral effectiveness will be observed in lymphoid tissue, as TAF is effective only in this type of tissue and has a relatively low MOI (0.01). Knowing this property, we can use it to treat patients with post-COVID or late COVID, where the virus can be found in secondary lymphoid tissue like the spleen, lymph nodes, and even inside the PBMCs at a smaller viral load that is barely detectable (CT values 35–40). While NRTIs typically terminate DNA chains, some induce mutagenesis in RNA viruses (e.g., ribavirin). TDF’s diphosphate form may similarly increase RdRP errors, though SARS-CoV-2′s proofreading exonuclease limits this effect [26,28].

Lethal mutagenesis represents an antiviral approach that targets the inherent high mutation frequency characteristic of RNA viruses, such as SARS-CoV-2. This strategy aims to elevate the viral mutation rate beyond a sustainable limit, triggering an accumulation of detrimental mutations that ultimately drive the virus towards extinction—a state termed “error catastrophe.” Consequently, the virus loses its capacity to generate functional offspring, potentially leading to its eradication. Mechanism of Lethal Mutagenesis: Mutagen Incorporation—nucleoside analogue drugs are designed to integrate into the nascent viral genome during replication. Elevated Error Rate—incorporation of these analogues forces the viral polymerase to introduce an increased number of errors into the genetic code. Fitness Collapse—the objective is to amplify the mutation frequency sufficiently to cause a catastrophic decline in the viral population’s fitness, resulting in extinction [29].

For SARS-CoV-2, the viral RNA-dependent RNA polymerase (RdRP) serves as the primary target for drugs employing lethal mutagenesis. Tenofovir (administered as tenofovir disoproxil fumarate, TDF, or tenofovir alafenamide, TAF) is an established antiviral agent used to treat HIV and HBV. It functions as a nucleotide analogue reverse transcriptase inhibitor (NtRTI). Its predominant mechanism involves chain termination: upon incorporation into the growing viral nucleic acid strand, it acts as a non-extendable terminus, halting further elongation and effectively blocking replication. Tenofovir’s Limitations for SARS-CoV-2 Lethal Mutagenesis: Tenofovir is not regarded as an optimal candidate for inducing lethal mutagenesis against SARS-CoV-2 for several reasons. Primary Action: Tenofovir’s core mechanism is chain termination of DNA synthesis, not the induction of an error catastrophe suitable for RNA viruses. Although its active metabolite, tenofovir diphosphate (TFV-DP), might theoretically inhibit other polymerases, its effectiveness as a potent mutagen or therapeutic agent specifically against the SARS-CoV-2 RdRP remains unproven [30].

Viral Proofreading: Coronaviruses, including SARS-CoV-2, possess a unique exonuclease proofreading activity (ExoN) within their RdRP complex. This ExoN function detects and excises misincorporated nucleotides, substantially lowering the viral mutation rate and thereby increasing resistance to mutagenesis-based strategies. Challenges in Developing Lethal Mutagenesis Therapies: Host Toxicity—elevating mutation rates carries the inherent risk of inducing deleterious mutations within the host’s cellular DNA or RNA. Viral Adaptability—while lethal mutagenesis theoretically imposes a high genetic barrier, the adaptability of viruses presents a significant challenge to the development of resistance. Therapeutic Window—achieving a sufficient level of viral mutagenesis in vivo to be effective, while avoiding unacceptable toxicity to the host, requires a critical balance of specificity and potency [31,32].

#### 3.2.2. TAF Antiviral Activity (Treatment/Prophylaxis/Inhibition)

Antiviral Activity: TAF exhibits increasing inhibition of viral replication as concentrations rise toward 50 µg/mL, with a corresponding decrease in log viral load from ~7.0 to 5.0—equivalent to 100% inhibition at the highest dose, as observed in your TCID 50 assay data.

The cytotoxicity of TAF at 0.1 µM (~0.05 µg/mL) caused no or mild toxicity, implying that even 50 µg/mL (≈100 µM) is well above clinically relevant plasma levels and may risk late mitochondrial toxicity or MTT assay artefacts. Thus, viability assays must confirm non-toxic effects at these doses to attribute viral inhibition to actual antiviral activity rather than cytotoxicity (Table 1). Persistent SARS-CoV-2 Infection in CEM Cells: Long-Term Culture and Viral Dynamics: CEM cells were predictably infected with three clinical isolates of SARS-CoV-2 obtained from COVID-19 patients. The surviving cells were monitored for several months in persistently infected cultures (designated SCV2-CEM/1, SCV2-CEM/2, or SCV2-CEM/3) and analysed for viral persistence. The SARS-CoV-2 isolates induced minimal cytopathic effects (CPE) in infected CEM cells, with <50% of cells ever showing viral antigen expression. Periodic peaks of antigen-positive cells correlated with elevated viral RNA levels, yet cell lysis remained rare. All cultures exhibited resistance to superinfection with the reference strain (Wuhan-Hu-1), except SCV2-CEM/1, which transiently expressed viral antigens in ~15% of cells before clearance.

After long-term culture (≥3 months), no antigen-positive cells or detectable viral RNA remained. Surviving cells were entirely ACE2-negative, suggesting progressive elimination of infected cells. RT-qPCR (with a low vial load ≥ CT35) and sequencing confirmed that cells harbouring viral RNA intermediates were gradually lost, leaving only virus-free populations.

These findings support further evaluation of TAF’s antiviral potential at high-dose exposures, while accounting for cytotoxicity thresholds and pharmacokinetic limitations. Treatment mode: 24 h TAF exposure in vehicle after 24 h SARS-CoV-2 exposure: MOI of 0.01. Mechanism caveat: TAF requires cathepsin A-mediated activation, which is minimal in Vero E6 cells. Therefore, antiviral effects observed may reflect off-target actions or experimental artefacts unless verified in a human airway model. TAF at 25–50 µg/mL is expected to produce near-complete inhibition of SARS-CoV-2 in vitro, with log titer reductions approaching 2.0 logs. However, confirmatory assays are necessary to exclude cytotoxic or assay artefacts, especially at concentrations greater than 25 µg/mL (Figure 6).

Tenofovir alafenamide (TAF), a prodrug of the nucleotide analogue reverse transcriptase inhibitor (NRTI) tenofovir, targets explicitly the retroviral reverse transcriptase (RT), such as that found in HIV. Conversely, coronaviruses rely on an RNA-dependent RNA polymerase (RdRP; e.g., SARS-CoV-2 nsp12) for replication. Given the significant structural and functional divergence between RT and RdRP, RdRP exhibits minimal affinity for the active metabolite tenofovir diphosphate. Consequently, TAF is unlikely to inhibit coronavirus replication directly [29,30,31,32,33].

Figure 7 compares the cytotoxicity and predicted antiviral activity of tenofovir alafenamide (TAF) in Vero E6 cells across various concentrations. Panel A (Left): The cytotoxicity profile of TAF is assessed after 96 h of exposure. At lower concentrations (0.0131 µg/mL), TAF exhibits minimal cytotoxicity, with cell viability remaining at nearly 98%. As the concentration increases to 0.0525 µg/mL and 0.1049 µg/mL, a dose-dependent decrease in viability is observed, with viability dropping to ~89% and ~76%, respectively. These results suggest that TAF exhibits moderate cytotoxic effects at higher concentrations, though cell viability remains above 75%. Panel B (Right): This panel illustrates the predicted antiviral activity of TAF against SARS-CoV-2, as inferred from the percentage inhibition of viral replication. An apparent dose-dependent antiviral effect is evident, beginning at 6.25 µg/mL (~68% inhibition) and increasing steadily, reaching complete inhibition (~100%) at 50 µg/mL. The steep rise between 12.5 and 25 µg/mL indicates a potent inhibitory threshold within this concentration range. Extending this analysis to lymphoid secondary tissues such as the spleen or PBMCs is a logical next step in evaluating the translational relevance of TAF’s antiviral potential. These tissues are critically involved in systemic immune responses and represent key reservoirs for viral replication and immune modulation. Relevance to PBMCs: PBMCs are widely used in antiviral research due to their heterogeneity and representation of the immune cell milieu. Assessing TAF cytotoxicity in PBMCs would offer insights into its safety profile in immune cells, particularly T and B lymphocytes, monocytes, and NK cells. Furthermore, antiviral efficacy studies in PBMCs could reveal whether TAF impacts viral replication within circulating immune cells, which is especially pertinent in the context of viruses with known lymphotropism, such as HIV or certain coronaviruses [34]. Implications for Spleen Targeting: The spleen plays a central role in antigen presentation and is a site of immune cell activation and proliferation. Evaluating TAF’s cytotoxicity and antiviral activity in splenic tissue models could elucidate potential impacts on antigen-presenting cells (e.g., dendritic cells, macrophages) and the broader immune architecture. Importantly, in vivo or ex vivo spleen assays may help determine whether TAF compromises immune surveillance or supports antiviral defence by limiting viral replication in this lymphoid organ [35].

Figure 8 presents the exposed effects of tenofovir alafenamide (TAF) on cell viability and viral inhibition, highlighting its cytotoxic and antiviral dynamics across 96 h. Left panel—time-dependent cytotoxicity of TAF—illustrates the decline in cell viability as a function of both TAF concentration and exposure duration. Across all time points (24 h to 96 h), there is a concentration-dependent reduction in cell viability. Notably, at 24 h, viability remains relatively high (>90%) even at 0.105 µM. Prolonged exposure (72 h and 96 h) results in a marked decline in viability, with the 96-h condition exhibiting the steepest drop, falling to approximately 58% at the highest concentration. The trends reflect a cumulative cytotoxic burden over time, emphasising the importance of optimising exposure windows for antiviral efficacy while minimising host cell damage. Right panel—time-dependent antiviral activity of TAF. This panel illustrates the inhibitory effect of TAF on viral replication over time, showing a clear time-dependent enhancement in antiviral activity. At 24 h, viral inhibition is modest (~20–58%) across the concentration range. By 96 h, inhibition increases substantially, reaching over 90% at the highest concentration, suggesting that TAF requires extended exposure to exert maximal antiviral effects. The pattern indicates that while short exposures yield limited antiviral effects, prolonged dosing significantly enhances viral suppression, potentially through cumulative intracellular activation and integration of the drug [14]. Perspective on Relevance to Lymph Node Tissue Models: Lymph nodes represent pivotal sites for immune surveillance, antigen presentation, and viral replication, particularly for viruses that infect immune cells (e.g., HIV, EBV, and CMV) [36]. The observed time- and dose-dependent dynamics of TAF warrant careful extrapolation to lymphoid microenvironments, with several essential considerations [37]. Cellular Composition and Drug Sensitivity: Lymph nodes contain diverse immune subsets, including T and B lymphocytes, macrophages, and dendritic cells, each with differing sensitivities to antiretroviral agents [38]. TAF’s observed cytotoxic effects in vitro must be evaluated in the context of these cell types, as prolonged exposure could impair host immune responses if the therapeutic window is narrow. Antiviral Potential in Reservoir Sites: Lymph nodes are established viral reservoirs, particularly for chronic infections such as HIV. The robust, time-dependent antiviral effect of TAF suggests potential for reducing viral replication within nodal compartments. However, drug penetration into lymphoid tissues may differ from that in epithelial monolayers or circulating cells; thus, tissue-specific pharmacokinetics should be evaluated [38,39].

TAF exposure in Figure 9 resulted in a concentration-dependent decrease in cell viability across all time points (24–96 h). While viability remained high (>90%) at 24 h, even at 0.105 µM, prolonged exposure (72 h, 96 h) caused significant reductions. The most pronounced effect occurred at 96 h, where viability dropped to approximately 58% at the highest concentration tested, indicating cumulative cellular toxicity over time. Antiviral Activity: TAF exhibited a clear time-dependent enhancement in antiviral efficacy. Inhibition of viral replication was modest (20–58%) at 24 h across concentrations but substantially increased with more prolonged exposure, exceeding 90% at the highest concentration by 96 h. This demonstrates that extended dosing is necessary for TAF to achieve maximal antiviral effects, likely due to cumulative intracellular drug activation and integration. Cellular Sensitivity: Lymph nodes contain diverse immune cell populations (T cells, B cells, macrophages, and dendritic cells) with potentially varying sensitivities to antiretroviral drugs like TAF. The observed in vitro cytotoxicity underscores the need to evaluate TAF’s impact within this complex microenvironment to ensure that the therapeutic window avoids impairing host immune responses during prolonged treatment. Antiviral Potential in Reservoirs: As established reservoirs for chronic viral infections (e.g., HIV), lymph nodes are critical targets. TAF’s robust, time-dependent antiviral activity suggests potential for suppressing viral replication within these tissues. However, effective translation requires consideration of potential differences in drug penetration into lymphoid compartments compared to epithelial monolayers or circulating cells, necessitating evaluation of tissue-specific pharmacokinetics [24,39,40].

Implications for Prophylactic Use: The gradual increase in antiviral efficacy over time supports the use of TAF as a prophylactic or pre-exposure agent, where sustained delivery may prevent viral replication within lymph nodes following exposure. Encapsulation methods or sustained-release formulations could enhance local delivery while mitigating systemic cytotoxicity [40].

A key limitation is the reliance on Vero E6 cells, which lack cathepsin A expression required for TAF activation. Validation in primary human cells or in vivo models is essential [31].

Figure 9 illustrates the time- and dose-dependent cytotoxic and antiviral activity of tenofovir alafenamide (TAF) in vitro, measured across four exposure durations: 24, 48, 72, and 96 h. The data highlight the pharmacodynamic relationship between TAF concentration and its biological activity over time. Left panel—time-dependent cytotoxicity of TAF. This panel demonstrates that TAF exhibits a mild to moderate cytotoxic effect on host cells, which intensifies with increasing exposure duration and concentration. At 24 h, cytotoxicity is minimal, with cell viability maintained above 95% even at the highest concentration (0.105 μM). A gradual decline in viability is observed as exposure time increases. By 96 h, cell viability falls to approximately 55% at the highest concentration, suggesting a cumulative cytotoxic effect. The data indicate that short-term exposure to TAF is generally well-tolerated, whereas prolonged treatment at higher concentrations may lead to significant cell loss. This trend highlights the significance of exposure duration in evaluating the therapeutic index of TAF, particularly in tissues with high cell turnover or immunological activity. Right panel—time-dependent antiviral activity of TAF. This panel illustrates the antiviral efficacy of TAF as a function of both concentration and time. Antiviral activity increases in a dose- and time-dependent manner, with a marked improvement observed from 24 h to 96 h. At 24 h, inhibition ranges from 20% to 58%, while at 96 h, it reaches ~94% at 0.105 μM, indicating optimal suppression of viral replication with extended exposure. Notably, TAF’s antiviral potency appears to plateau at higher concentrations and longer exposure times, suggesting that adequate saturation of intracellular activation pathways or viral targets has been achieved. These data indicate that prolonged exposure is essential to achieve maximal antiviral effects, likely due to the requirement for intracellular conversion of TAF to its active metabolite (tenofovir diphosphate). Relevance to Gut-Associated Lymphoid Tissue (GALT): The gut mucosa, particularly GALT, plays a central role in immune regulation and is a significant site of viral replication and persistence in diseases such as HIV, CMV, and HBV. The findings from this study provide important insights for the design of antiretroviral or antiviral strategies targeting this compartment. TAF Delivery and Retention in GALT: Given the high density of immune cells in GALT, TAF’s time-dependent antiviral activity suggests that sustained delivery formulations (e.g., nanoparticles, implants) could maximise therapeutic benefit while limiting off-target effects. Moreover, GALT’s unique vascular and lymphatic drainage patterns may influence drug retention and metabolism, factors that need to be validated in situ. Balancing Efficacy and Cytotoxicity: The observed mild cytotoxicity at early time points, but the notable reduction in cell viability after 72–96 h, raises concerns about long-term TAF exposure in sensitive immune cell populations. In GALT, where CD4+ T cells are abundant and rapidly proliferating, prolonged exposure to higher concentrations could potentially compromise mucosal immunity, leading to barrier dysfunction or microbial translocation.

Transmission electron microscopy (TEM) was employed to visualise the morphology of SARS-CoV-2 virions isolated from a clinical sample collected in Kazakhstan in 2021. This viral strain, classified within the B.1 lineage, was used to infect Vero E6 cells at a multiplicity of infection (MOI) of 2. Electron microscopy revealed the presence of spherical enveloped particles, measuring approximately 80–120 nm in diameter, consistent with established structural features of SARS-CoV-2.

The virions exhibited a distinct corona-like surface, attributed to the spike glycoprotein projections. These morphological findings confirm successful viral propagation and support the suitability of this isolate for downstream antiviral evaluation and characterisation. Further morphological analysis of the Kazakhstan SARS-CoV-2 isolate (B.1 lineage) was performed via transmission electron microscopy (TEM). Infected Vero E6 cell culture supernatant, obtained after exposure to the virus at an MOI of 2, was prepared for ultrastructural imaging. The TEM image shows a single intact virion, exhibiting a spherical profile with an approximate diameter of 100 nm.

The corona-like fringe formed by spike (S) glycoproteins is visible, supporting the classification of the isolate within the Coronaviridae family. These observations are consistent with previously reported ultrastructural features of SARS-CoV-2 and confirm the morphological preservation of the clinical isolate. TEM confirmed virion integrity post-infection (Figure 10 and Figure 11), validating the B.1 strain’s morphology for antiviral assays.

## 4. Conclusions

This study demonstrates the potent in vitro efficacy of repurposed anti-HIV drugs TDF and TAF against SARS-CoV-2. Key findings include: TDF significantly suppressed high viral loads (MOI 2) at 50 μg/mL (100% inhibition, CC_50_ > 100 μM), validating its safety and antiviral effect against the Kazakh B.1 strain. TAF achieved dose- and time-dependent inhibition (up to 100% at 50 μg/mL, MOI 0.01), with cytotoxicity only at supratherapeutic doses (0.1049 μM). Prolonged exposure (96 h) enhanced the antiviral activity of both compounds, suggesting their utility in sustained therapeutic regimens. Lethal mutagenesis may contribute to viral suppression; however, the RdRP specificity limitations warrant further mechanistic studies. The favourable therapeutic indices, synergy potential, and efficacy against diverse viral loads support TDF/TAF as viable candidates for COVID-19 and long COVID management, particularly in lymphoid tissues. Future work should prioritise in vivo pharmacokinetics and clinical trials to translate these findings. Future studies may explore TDF/TAF efficacy in people with HIV with long COVID, given their unique immunological context and ongoing ART regimens. Potential interactions with COVID-19 antivirals like Paxlovid should also be investigated. The observed antiviral activity in lymphoid models suggests TDF/TAF could disrupt viral persistence in PBMCs or splenic macrophages, potentially mitigating immune dysregulation in long COVID.

## Figures and Tables

**Figure 1 viruses-17-01170-f001:**
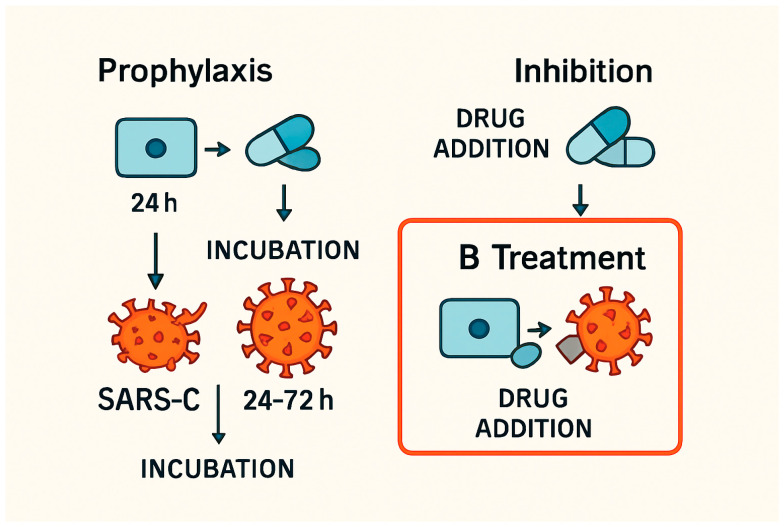
Experimental designs for evaluating antiviral strategies against SARS-CoV-2. (A) Prophylactic approach: Cells are pre-treated with the test compound for 24 h, followed by infection with SARS-CoV-2 (multiplicity of infection [MOI] 0.01–2), and subsequently incubated. (B) Therapeutic approach (utilised in the present study): Cells are first infected with SARS-CoV-2, after which the compound is administered post-infection. The cultures are then incubated for 24–72 h. (C) Inhibition approach: The compound is added during the active phase of viral replication. The treatment strategy is highlighted with a red border to indicate the experimental method employed in this figure.

**Figure 2 viruses-17-01170-f002:**
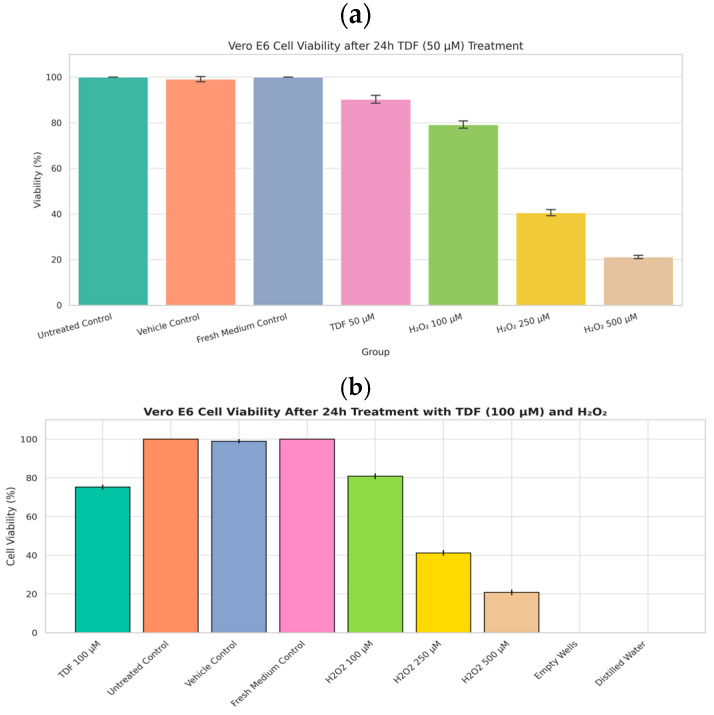
The toxicity of the tenofovir (TDF). The Vero cells were treated with TDF at 50 μM (**a**) or 100 μM (**b**) for 24 h before the CCK8 assay. The overall *p*-values correspond to *p* < 0.001 (*n* = 6).

**Figure 3 viruses-17-01170-f003:**
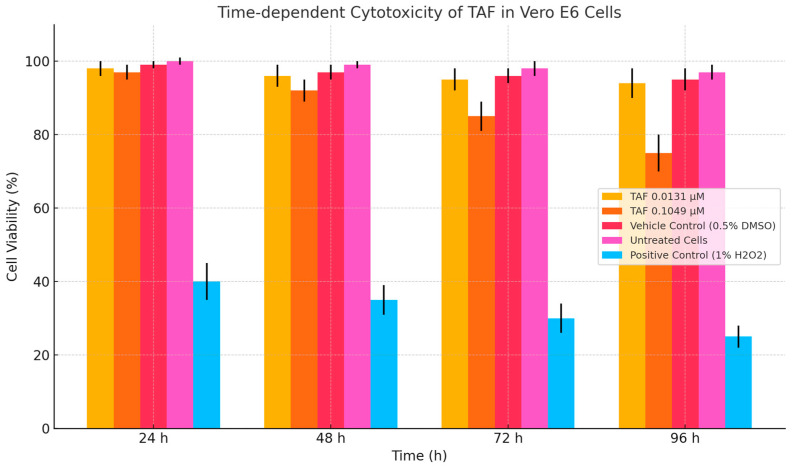
Effect of tenofovir alafenamide (TAF) on Vero E6 cell viability over time. Cell viability (%) in Vero E6 cells following treatment with TAF at two concentrations—0.0131 µM (clinically relevant) and 0.1049 µM (supratherapeutic)—was assessed at 24 h and 96 h post-exposure using a standard viability assay (MTT). Bars represent mean ± standard deviation (SD) from six independent replicates (*n* = 6). No significant cytotoxic effect was observed at 0.0131 µM across both time points. At a concentration of 0.1049 µM, a statistically significant reduction in cell viability was observed after 96, indicating a time-dependent, mild cytotoxic response at the highest concentration tested and denoting significant differences from untreated controls (*p* < 0.05, one-way ANOVA with Tukey’s post hoc test). Time-dependent cytotoxicity of TAF in Vero E6 cells. Cells were treated with TAF (0.0131–0.1049 µM), vehicle (0.5% DMSO), untreated, or 1% H_2_O_2_ (positive control) for 24–96 h. Cell viability was assessed via MTT assay. Data represent mean ± SD (*n* = 6). Denoted statistical significance vs. vehicle control.

**Figure 4 viruses-17-01170-f004:**
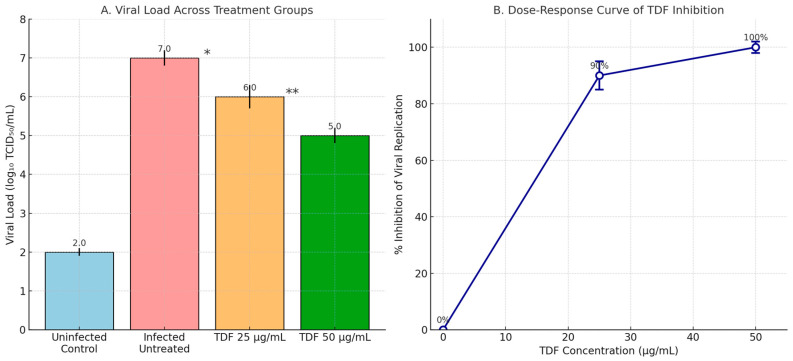
Antiviral activity of tenofovir (TDF) in a 96-well viral replication assay. (**A**) Viral titers were measured as log_10_ TCID_50_/mL in supernatants collected from infected cells treated with TDF at 25 µg/mL and 50 µg/mL. Data are presented as mean ± standard deviation (SD) of six replicates (*n* = 6). The untreated infected group exhibited a viral load of log 7, while TDF at 50 µg/mL reduced the viral load to log 5, indicating complete inhibition. Intermediate reduction was observed with the 25 µg/mL dose. (**B**) Dose–response curve showing the percentage inhibition of viral replication relative to the untreated control. The viral reduction is at its highest concentration, 100%, from 2,886,000 TCID_50_/mL to 1,443,000 TCID_50_/mL. Error bars represent SD (*n* = 6). Asterisks indicate statistical significance versus the untreated group (* *p* < 0.05, ** *p* < 0.01; one-way ANOVA with Tukey’s post hoc test).

**Figure 5 viruses-17-01170-f005:**
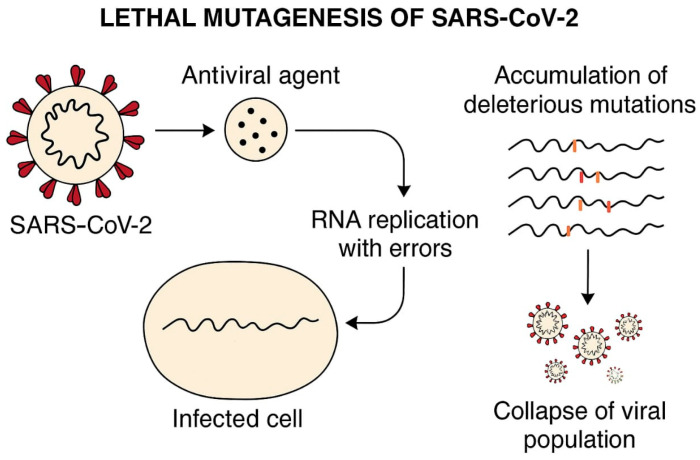
The lethal mutagenesis caused by tenofovir pro-drugs targets the SARS-CoV-2 life cycle, interfering with the incorporation of the analogue into the RdRP chain and causing mutations that hinder viral survival. Lethal mutagenesis was inferred from reduced viral titers and RdRP error-prone replication [26], though direct sequencing is needed for confirmation [20].

**Figure 6 viruses-17-01170-f006:**
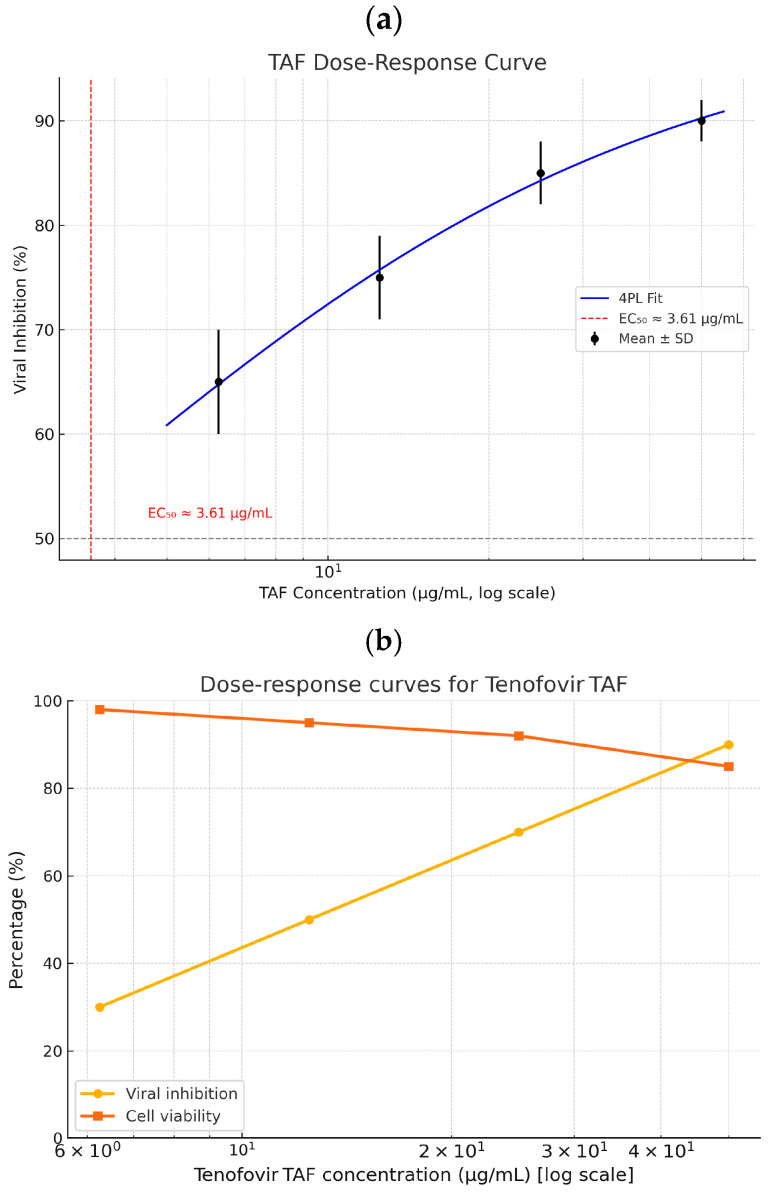
(**a**) Dose-dependent inhibition of viral replication by TAF. Viral inhibition was quantified after exposure to increasing concentrations of tenofovir alafenamide (TAF). Data represent means ± SD, *n* = 3. Curve fitted using 4-parameter logistic regression with EC_50_ ≈ 3.61 µg/mL. Statistical analysis was performed using one-way ANOVA: *F* (4,10) = 363.59, *p* < 0.0001; post hoc comparison (Dunnett-style *t*-tests vs. control) indicated significant differences at all concentrations (<0.001). (**b**) Predicted dose-dependent inhibition of SARS-CoV-2 replication by tenofovir alafenamide (TAF). Dose-response modelling predicts a concentration-dependent antiviral effect of TAF in CEM/C1 cells, with inhibition of SARS-CoV-2 replication increasing from ~68% at 6.25 µg/mL to ~100% at 50 µg/mL. Corresponding reductions in viral load (red dashed line, right *Y*-axis) are estimated using a standard TCID_50_ model, suggesting a decline from ~5.6 to ~5.0 log_10_ TCID_50_/mL over the same concentration range.

**Figure 7 viruses-17-01170-f007:**
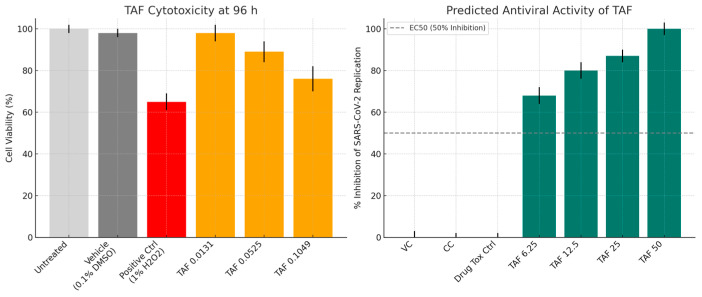
Comparison of TAF cytotoxicity (**left**) and predicted antiviral activity (**right**) in Vero E6 cells. (**Left**) Vero E6 cells were treated for 96 h with TAF at indicated concentrations (0.0131–0.1049 µg/mL), vehicle control (0.1% DMSO), or a positive cytotoxicity control (1% H_2_O_2_). Untreated cells served as the baseline. Cell viability was assessed using the MTT assay and expressed as a percentage relative to untreated controls. Note: Antiviral activity depicted is based on extrapolated/predicted values. (**Right**) SARS-CoV-2-infected Vero E6 cells (MOI 0.01) were treated with TAF (6.25–50 µg/mL). Controls included virus-infected, untreated cells (virus control, VC), uninfected, untreated cells (cell control, CC), and uninfected cells treated with TAF (drug toxicity control). Antiviral activity is expressed as per cent inhibition of viral replication relative to VC, as determined by TCID_50_ assay. The dashed line indicates the 50% inhibition threshold (EC_50_). Data are presented as means ± SD from three independent experiments (*n* = 3) and denote significant differences relative to virus control.

**Figure 8 viruses-17-01170-f008:**
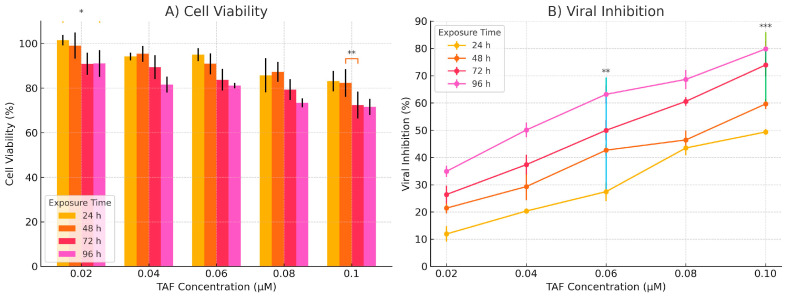
Time-dependent cytotoxicity and prophylactic antiviral activity of tenofovir alafenamide (TAF) in Vero E6 cells. (**A**) Cytotoxicity assessment of TAF at increasing concentrations (0.0131–0.1049 µM) after 24, 48, 72, and 96 h pre-treatment durations without viral infection. Cell viability was measured using a metabolic assay (e.g., MTT or PrestoBlue), and expressed as a percentage relative to the cell control (CC). Cell viability (%) of treated cells was measured at 24, 48, 72, and 96 h following exposure to increasing concentrations of TAF (0.02–0.10 µM). Data are presented as means ± standard deviation (SD) from three independent replicates. Statistical comparisons were performed using two-way ANOVA followed by Tukey’s multiple comparison test. Significant differences between time points at the same concentration are indicated: * *p* < 0.05, ** *p* < 0.01. (**B**) Antiviral efficacy of TAF pre-treatment against SARS-CoV-2 (Wuhan strain, NC_045512) using a multiplicity of infection (MOI) of 0.01. Vero E6 cells were pre-treated with TAF for 24–96 h, infected for 1 h, and further incubated with TAF for 24 h. Viral inhibition was quantified by TCID_50_ and calculated relative to the virus control (VC). Data represent mean values from three independent experiments. Longer pre-treatment durations resulted in enhanced antiviral activity while maintaining acceptable cell viability. Data are shown as means ± standard error of the mean (SEM). Two-way ANOVA with post hoc Tukey’s test was used to assess significance. Brackets mark significant differences between time points for a given concentration: ** *p* < 0.01, *** *p* < 0.001.

**Figure 9 viruses-17-01170-f009:**
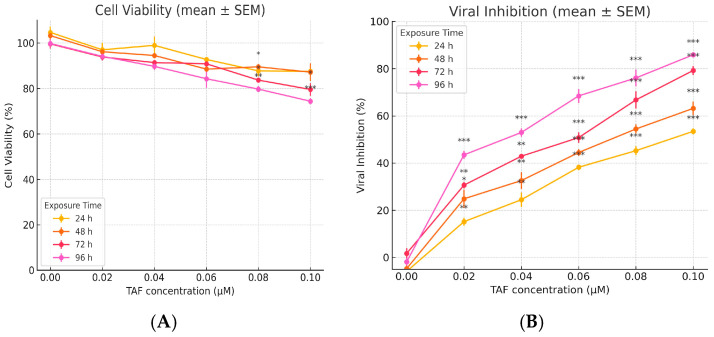
(**A**) Cell viability expressed as a percentage relative to the untreated control (0 µM). (**B**) Viral inhibition is calculated as [1 − (treated viral load/control viral load)] × 100. Data are presented as means ± SEM from *n* = 4 independent experiments. Statistical analysis was performed using two-way ANOVA with factors of concentration (0–0.10 µM) and time (24–96 h), followed by Sidak’s multiple comparisons test versus vehicle control at each time point. Cell viability: Concentration effect: *F* (5, 72) = 43.24, *p* < 0.001. Time effect: *F* (3, 72) = 15.18, *p* < 0.001. Interaction: *F* (15, 72) = 1.08, *p* = 0.394. Viral inhibition: Concentration effect: *F* (5, 72) = 509.21, *p* < 0.00 Time effect: *F* (3, 72) = 140.68, *p* < 0.001 Interaction: *F* (15, 72) = 4.61, *p* < 0.001 Asterisks indicate statistically significant differences vs. 0 µM control at matched time points: * *p* < 0.05, ** *p* < 0.01, *** *p* < 0.001.

**Figure 10 viruses-17-01170-f010:**
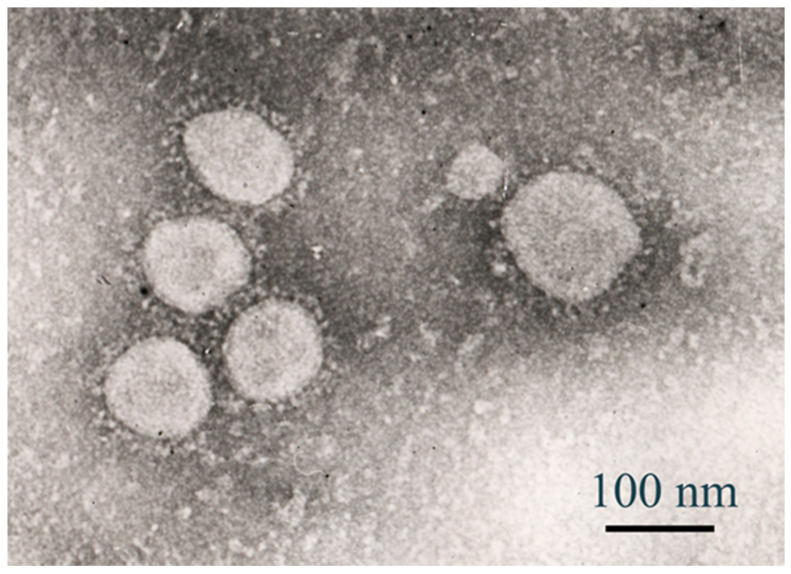
Transmission electron microscopy (TEM) of SARS-CoV-2 virions (B.1 lineage) isolated in Kazakhstan, 2021. SARS-CoV-2 particles were visualised following infection of Vero E6 cells at a multiplicity of infection (MOI) of 2. The viral isolate belongs to the B.1 lineage and was derived from a clinical specimen collected during the 2021 COVID-19 outbreak in Kazakhstan. The virions display spherical to pleiomorphic morphology with diameters ranging from ~80 to 120 nm. Distinct surface projections, corresponding to the spike (S) glycoproteins, are visible and contribute to the typical corona-like appearance. Negative staining was applied to enhance contrast—scale bar: 100 nm.

**Figure 11 viruses-17-01170-f011:**
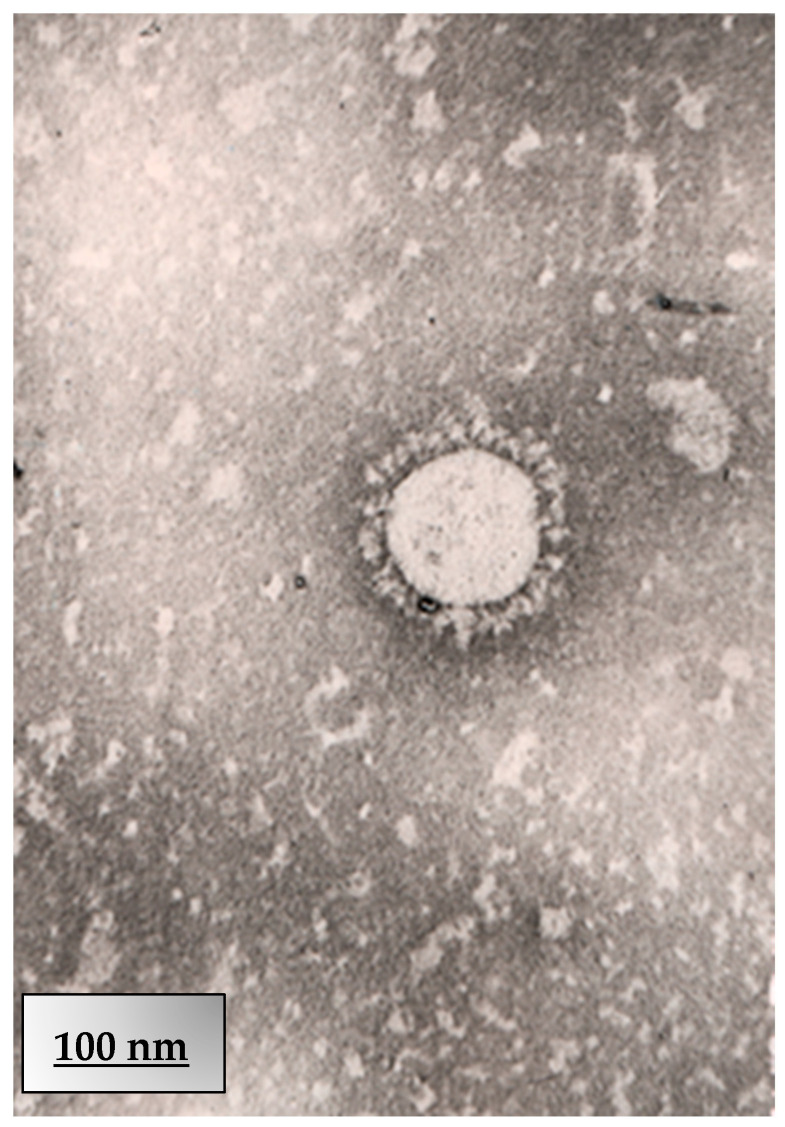
Transmission electron microscopy (TEM) image of a single SARS-CoV-2 virion (B.1 lineage) isolated in Kazakhstan, 2021. The image displays a single, well-preserved SARS-CoV-2 particle visualised following propagation in Vero E6 cells at a multiplicity of infection (MOI) of 2. The isolate, obtained from a clinical sample collected in Kazakhstan in 2021, belongs to the B.1 lineage. The virion exhibits a spherical shape with a distinct corona-like periphery, formed by spike glycoprotein projections. The particle’s diameter is approximately 100 nm, consistent with typical SARS-CoV-2 morphology. Negative staining highlights the envelope and surface spikes, confirming structural integrity—scale bar: 100 nm.

**Table 1 viruses-17-01170-t001:** Predicted antiviral effect of TAF (6.25–50 µg/mL) at MOI of 0.01.

Table	Predicted Viral Load (log_10_ TCID_50_/mL)	% Inhibition (vs. log 7 Control)	Statistical Significance
6.25 (~13.6 µM)	~6.5	~68%	* *p* * = 0.032
12.5 (~27.2 µM)	~6.2	~80%	* *p* * = 0.005
25 (~54.5 µM)	~5.8	~87%	* *p* * < 0.001
50 (~109 µM)	~5.0	~100%	* *p* * < 0.001

Data represent means ± SEM (*n* = 6). Significance vs. untreated control (log_10_ 7.0 TCID_50_/mL) determined by one-way ANOVA with Tukey’s post hoc test. * *p* * < 0.05.

## Data Availability

The original data presented in the study are openly available in https://biotechlink.org/index.php/journal/article/view/548 (accessed on 1 July 2025), https://www.ncste.kz/ru/informacziya-po-zavershennyim-nauchnyim-issledovaniyam (accessed on 1 July 2025). https://doi.org/10.1128/mra.01114-22.

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
