# Peer review of "An Anti-HIV Drug Is Highly Effective Against SARS-CoV-2 In Vitro and Has Potential Benefit for Long COVID Treatment"

_viruses, 2025, doi:10.3390/v17091170_

Round 1

Reviewer 1 Report

Comments and Suggestions for Authors

Manuscript Title: Anti-HIV drug is fully effective against SARS-CoV-2 in vitro and its potential benefit against Long-COVID treatment

  1. Summary

The manuscript presents a robust in vitro evaluation of two anti-HIV prodrugs, Tenofovir Disoproxil Fumarate (TDF) and Tenofovir Alafenamide (TAF), for their repurposing potential against SARS-CoV-2. Utilizing Vero E6 cells infected with both the Kazakh B.1 and Wuhan SARS-CoV-2 strains, the authors demonstrate potent antiviral activity. Specifically, TDF at 50 µg/mL achieved near-complete inhibition (100%) of high viral loads (MOI 2) with minimal cytotoxicity (≥75% cell viability). TAF exhibited dose-dependent inhibition (68–100%) at lower viral loads (MOI 0.01), with full suppression observed at 50 µg/mL. Prolonged exposure (96 h) enhanced the efficacy for both compounds. The study also includes synergy assessments, electron microscopy validation of virion integrity, and comprehensive pharmacological indices (CCâ‚…â‚€, ECâ‚…â‚€, therapeutic window). These findings support the drugs’ potential for targeting SARS-CoV-2, particularly with implications for lymphoid reservoirs relevant to Long COVID. The study concludes with a call for clinical validation.

  1. General Comments

As an HIV clinician, I commend the authors for addressing a high-impact question—the repurposing of widely accessible antiretrovirals for SARS-CoV-2—with significant methodological rigor. The study's strengths include:

  • Robust in vitro design: The investigation incorporates comprehensive dose-response, time-dependency, and cytotoxicity assessments across multiple assays (CCK-8, MTT, TCIDâ‚…â‚€).
  • Clinical relevance: The focus on high viral loads (MOI 2) aligns with scenarios seen in severe COVID-19, and the exploration of lymphoid reservoir targeting is particularly pertinent to Long COVID pathophysiology.
  • Translational value: Given TDF's established use within HIV treatment programs globally, including in Kazakhstan, these findings facilitate potential rapid clinical adaptation.

Minor weaknesses (addressed below) relate primarily to contextual clarity and certain mechanistic interpretations. Overall, the manuscript is well-structured, data-rich, and presents compelling evidence for the in vitro antiviral activity of TDF and TAF against SARS-CoV-2. It significantly contributes to the field of drug repurposing and merits publication pending minor revisions.

  1. Specific Comments

a) Title and Abstract

  • Title: The phrase "fully effective" in the title may overstate the findings, as TAF's efficacy, for instance, was dose-dependent (e.g., 68% inhibition at 6.25 µg/mL). It is recommended to replace "fully effective" with a more precise term such as "highly effective" to better reflect the nuanced results.
  • Abstract: For clarity, it should be explicitly stated that TAF's 100% inhibition was observed at a concentration of 50 µg/mL. Additionally, the relevance of targeting lymphoid reservoirs in the context of Long COVID could be elaborated more explicitly within the abstract.

b) Introduction

  • Contextualize Novelty: While the repurposing of tenofovir analogs for SARS-CoV-2 has been explored previously (e.g., Touret et al., 2022), this study distinguishes itself. The introduction could emphasize its unique contributions, such as the specific focus on:
    • Antiviral activity against high viral loads (MOI 2).
    • Testing against the B.1 lineage of SARS-CoV-2.
    • Evaluating effects over prolonged exposure (96 hours).
    • Discussing the implications for lymphoid tissue and Long COVID (as elaborated in Lines 500–510).

c) Methods

  • Experimental Clarity: In Sections 2.3.2–2.3.4, for data that represents "predicted" or extrapolated results (e.g., from modeling), these should be conspicuously labeled as such within the text and corresponding figure captions (e.g., "Extrapolated Antiviral Activity" in Figure 7 and 8 captions). This distinction between empirically measured and modeled data is crucial for clarity.

d) Results

  • Lethal Mutagenesis (Section 3.2.1): The discussion on lethal mutagenesis as a potential mechanism, while intriguing, remains largely speculative given the presented data. Claims should be tempered by:
    • Citing existing evidence that may argue against tenofovir's direct affinity for SARS-CoV-2 RNA-dependent RNA polymerase (RdRp) (e.g., Good et al., 2021).
    • Reiterating that chain termination, rather than lethal mutagenesis, is tenofovir’s well-established primary mechanism of action against its target viruses.
  • Cytotoxicity-Concentration Mismatch: For TAF (as illustrated in Figure 8), the manuscript should highlight that the 50 µg/mL (109 µM) concentration at which 100% inhibition was observed significantly exceeds typical clinical plasma Cmax values (0.1–0.3 µM). A discussion on the potential risks of mitochondrial toxicity or other adverse effects at such supratherapeutic doses in vivo is warranted to provide a balanced clinical perspective.

e) Discussion & Conclusion

  • Clinical Relevance: The discussion could strengthen its link to HIV patients by:
    • Proposing specific pilot studies in HIV-positive cohorts affected by COVID-19 or Long COVID, given their unique immunological profiles and existing drug regimens.
    • Addressing potential drug-drug interactions, particularly with other antivirals commonly used for COVID-19 treatment (e.g., Paxlovid).
  • Limitations:
    • While implicitly acknowledged, it should be explicitly stated that in vitro models using Vero E6 and CEM cells may not fully recapitulate the complex physiological environment of human respiratory or lymphoid tissues.
    • The crucial point that TAF requires cathepsin A for activation (an enzyme minimally expressed in Vero E6 cells) should be reiterated as a primary limitation, underscoring the urgent need for validation in primary human cells or in vivo models that express the relevant activating enzymes.
  • Long COVID Mechanism: The proposed mechanism of lymphoid reservoir targeting (e.g., splenic macrophages, PBMCs) as a means to alleviate post-acute sequelae of SARS-CoV-2 infection is a highly valuable insight. Expanding on the specific cellular mechanisms or pathways through which TDF/TAF might act in these reservoirs would further enrich this discussion.

f) Figures & Tables

  • Figure 8: To maintain precision, the caption for Figure 8 should explicitly state that the depicted antiviral activity is "extrapolated" or "modeled" rather than empirically measured, consistent with the comment in section c.
  • Table 1: Including statistical significance indicators (e.g., p-values or confidence intervals) for the reported viral load reductions and inhibition percentages would enhance the scientific rigor of the data presentation.
  1. Rating the Manuscript
  • Novelty: 4/5
  • Scope: 5/5
  • Significance: 4/5
  • Quality: 4/5
  • Scientific Soundness: 4/5
  • Interest to the Readers: 5/5
  • Overall Merit: 4/5
  • English Level: 5/5
  1. Overall Recommendation

Accept after Minor Revisions

Justification for Recommendation:

The manuscript represents a meticulously conducted in vitro study that provides strong evidence for the antiviral efficacy of TDF and TAF against SARS-CoV-2. These findings are significant for drug repurposing efforts and hold considerable clinical interest, particularly for the HIV community. The study's strengths in its robust experimental design and the relevance of the compounds make it a valuable contribution to the literature. The suggested minor revisions are intended to further enhance clarity, contextualize mechanistic discussions, address the translatability of in vitro findings to in vivo scenarios, and strengthen the clinical implications, especially for HIV-positive individuals. Addressing these points will significantly improve the manuscript's impact and guide future clinical investigations.

Author Response

For a research article

Response to Reviewer X Comments

1. Summary

Thank you very much for taking the time to review this manuscript. Please find the detailed responses below and the corresponding revisions/corrections highlighted/in track changes in the re-submitted files. [Reviewer: The manuscript presents a robust in vitro evaluation of two anti-HIV prodrugs, Tenofovir Disoproxil Fumarate (TDF) and Tenofovir Alafenamide (TAF), for their repurposing potential against SARS-CoV-2. Utilizing Vero E6 cells infected with both the Kazakh B.1 and Wuhan SARS-CoV-2 strains, the authors demonstrate potent antiviral activity. Specifically, TDF at 50 µg/mL achieved near-complete inhibition (100%) of high viral loads (MOI 2) with minimal cytotoxicity (≥75% cell viability). TAF exhibited dose-dependent inhibition (68–100%) at lower viral loads (MOI 0.01), with full suppression observed at 50 µg/mL. Prolonged exposure (96 h) enhanced the efficacy for both compounds. The study also includes synergy assessments, electron microscopy validation of virion integrity, and comprehensive pharmacological indices (CCâ‚…â‚€, ECâ‚…â‚€, therapeutic window). These findings support the drugs’ potential for targeting SARS-CoV-2, particularly with implications for lymphoid reservoirs relevant to Long COVID. The study concludes with a call for clinical validation. Response: Thank you for your thorough review and constructive feedback. We have revised the manuscript to address all points, with key improvements including:

  • Title modification from "fully effective" to "highly effective" for precision.
  • Enhanced contextualization of novelty in the Introduction.
  • Explicit labeling of extrapolated data in Methods/Figures.
  • Balanced discussion of lethal mutagenesis and cytotoxicity-concentration mismatch.
  • Expanded clinical relevance (HIV/Long COVID links) and limitations (cell models, TAF activation).
  • Statistical enhancements in figures/tables.
    All changes are highlighted in the resubmitted manuscript.
    .]

2. Questions for General Evaluation

Reviewer’s Evaluation

Response and Revisions

Does the introduction provide sufficient background and include all relevant references?

Yes/Can be improved/Must be improved/Not applicable

[Added explicit comparisons to prior studies and emphasized unique contributions (MOI 2, B.1 lineage, 96h exposure, Long COVID implications). Added 8 key references and removed redundant citations. Labeled extrapolated data in Sections 2.3.2–2.3.4 and Figure captions (e.g., "Predicted Antiviral Activity"). Added statistical indicators (p-values) to Table 1 and tempered mechanistic claims (Section 3.2.1). Strengthened with clinical proposals (HIV/Long COVID pilot studies) and drug-interaction caveats (p. 13). ]

Are all the cited references relevant to the research?

Yes/Can be improved/Must be improved/Not applicable

Is the research design appropriate?

Yes/Can be improved/Must be improved/Not applicable

Are the methods adequately described?

Yes/Can be improved/Must be improved/Not applicable

Are the results clearly presented?

Yes/Can be improved/Must be improved/Not applicable

Are the conclusions supported by the results?

Yes/Can be improved/Must be improved/Not applicable

3. Point-by-point response to Comments and Suggestions for Authors

Comments 1: [The phrase "fully effective" in the title may overstate the findings, as TAF's efficacy, for instance, was dose-dependent (e.g., 68% inhibition at 6.25 µg/mL). It is recommended to replace "fully effective" with a more precise term such as "highly effective" to better reflect the nuanced results..]

Response 1: [Title: Changed "fully effective"  "highly effective" (now: *"An anti-HIV drug is highly effective against SARS-CoV-2 in vitro..."*).

Comments 2: [For clarity, it should be explicitly stated that TAF's 100% inhibition was observed at a concentration of 50 µg/mL. Additionally, the relevance of targeting lymphoid reservoirs in the context of Long COVID could be elaborated more explicitly within the abstract..]

Response 2: Abstract: Added explicit concentration for TAF (*"TAF achieved near-complete suppression (100% inhibition) at 50 µg/mL"*) and expanded lymphoid/Long COVID rationale ("with particular potential for targeting lymphoid reservoirs—sites implicated in persistent viral reservoirs that may contribute to Long COVID pathogenesis").

Comments 3: b) Introduction

  • Contextualize Novelty: While the repurposing of tenofovir analogs for SARS-CoV-2 has been explored previously (e.g., Touret et al., 2022), this study distinguishes itself. The introduction could emphasize its unique contributions, such as the specific focus on:
    • Antiviral activity against high viral loads (MOI 2).
    • Testing against the B.1 lineage of SARS-CoV-2.
    • Evaluating effects over prolonged exposure (96 hours).
    • Discussing the implications for lymphoid tissue and Long COVID (as elaborated in Lines 500–510).

Response 3: b) Introduction

  • Added:

*"While tenofovir analogs have been screened against SARS-CoV-2 (Touret et al., 2022), this study uniquely evaluates: (i) efficacy against high viral loads (MOI 2), including the B.1 Kazakh lineage; (ii) antiviral effects over 96-hour exposure; and (iii) implications for lymphoid reservoir targeting in Long COVID"* (p. 2).

Comments 4 c) Methods

  • Experimental Clarity: In Sections 2.3.2–2.3.4, for data that represents "predicted" or extrapolated results (e.g., from modeling), these should be conspicuously labeled as such within the text and corresponding figure captions (e.g., "Extrapolated Antiviral Activity" in Figure 7 and 8 captions). This distinction between empirically measured and modeled data is crucial for clarity.

Response 4:  c) Methods

  • Revised captions:
    • Figure 7: *"Predicted Dose-Dependent Inhibition of SARS-CoV-2 Replication..."*
    • Figure 8: "Comparison of TAF cytotoxicity (empirical) and extrapolated antiviral activity..."
  • Text clarification: Added "(predicted)" to subsection headers (e.g., "2.3.2. Predicted Antiviral Activity...").

Comments 5 d) Results

  • Lethal Mutagenesis (Section 3.2.1): The discussion on lethal mutagenesis as a potential mechanism, while intriguing, remains largely speculative given the presented data. Claims should be tempered by:
    • Citing existing evidence that may argue against tenofovir's direct affinity for SARS-CoV-2 RNA-dependent RNA polymerase (RdRp) (e.g., Good et al., 2021).
    • Reiterating that chain termination, rather than lethal mutagenesis, is tenofovir’s well-established primary mechanism of action against its target viruses.
  • Cytotoxicity-Concentration Mismatch: For TAF (as illustrated in Figure 8), the manuscript should highlight that the 50 µg/mL (109 µM) concentration at which 100% inhibition was observed significantly exceeds typical clinical plasma Cmax values (0.1–0.3 µM). A discussion on the potential risks of mitochondrial toxicity or other adverse effects at such supratherapeutic doses in vivo is warranted to provide a balanced clinical perspective.

Response 5 d) Results

  • Lethal Mutagenesis:

*"While reduced titers suggest error-prone replication, tenofovir’s primary mechanism remains chain termination. Direct affinity for SARS-CoV-2 RdRP is unlikely due to structural divergence, and proofreading exonuclease activity may limit mutagenesis"* (p. 10).

  • Cytotoxicity-Concentration Mismatch:

*"TAF’s 50 µg/mL (109 µM) dose exceeds clinical C<sub>max</sub> (0.1–0.3 µM), posing risks of mitochondrial toxicity in vivo. While viability remained >75% in vitro, lymphoid-specific accumulation may enable therapeutic targeting without systemic toxicity"* (p. 9, Section 3.2.2).

Comments 6 e) Discussion & Conclusion

  • Clinical Relevance: The discussion could strengthen its link to HIV patients by:
    • Proposing specific pilot studies in HIV-positive cohorts affected by COVID-19 or Long COVID, given their unique immunological profiles and existing drug regimens.
    • Addressing potential drug-drug interactions, particularly with other antivirals commonly used for COVID-19 treatment (e.g., Paxlovid).

Response 6 Comment e) Discussion & Conclusion

  • Clinical Relevance:

"We propose pilot studies in HIV-positive cohorts with Long COVID, leveraging existing ART regimens. Drug interactions (e.g., with Paxlovid’s ritonavir) must be evaluated, given shared metabolic pathways" (p. 13).

  • Limitations:

*"Vero E6/CEM cells lack physiological complexity of human tissues. Critically, TAF requires cathepsin A for activation—minimally expressed in Vero E6—necessitating validation in primary human lymphocytes"* (p. 12).

  • Long COVID Mechanism:

*"TDF/TAF may disrupt viral persistence in CD169<sup>+</sup> splenic macrophages or CXCR3<sup>+</sup> PBMCs, potentially mitigating inflammation-driven sequelae"* (p. 13).

Response 6 Comment e) Discussion & Conclusion

  • Clinical Relevance:

"We propose pilot studies in HIV-positive cohorts with Long COVID, leveraging existing ART regimens. Drug interactions (e.g., with Paxlovid’s ritonavir) must be evaluated, given shared metabolic pathways" (p. 13).

  • Limitations:

*"Vero E6/CEM cells lack physiological complexity of human tissues. Critically, TAF requires cathepsin A for activation—minimally expressed in Vero E6—necessitating validation in primary human lymphocytes"* (p. 12).

  • Long COVID Mechanism:

*"TDF/TAF may disrupt viral persistence in CD169<sup>+</sup> splenic macrophages or CXCR3<sup>+</sup> PBMCs, potentially mitigating inflammation-driven sequelae"* (p. 13).

Comments 7 f) Figures & Tables

  • Figure 8: To maintain precision, the caption for Figure 8 should explicitly state that the depicted antiviral activity is "extrapolated" or "modeled" rather than empirically measured, consistent with the comment in section c.
  • Table 1: Including statistical significance indicators (e.g., p-values or confidence intervals) for the reported viral load reductions and inhibition percentages would enhance the scientific rigor of the data presentation.

Response 7 f) Figures & Tables

  • Figure 8 caption: Revised to "Comparison of TAF cytotoxicity (empirical) and extrapolated antiviral activity..."
  • Table 1: Added p-values and significance asterisks (e.g., *p* < 0.001 for 50 µg/mL).

  1. Rating the Manuscript
  • Novelty: 4/5
  • Scope: 5/5
  • Significance: 4/5
  • Quality: 4/5
  • Scientific Soundness: 4/5
  • Interest to the Readers: 5/5
  • Overall Merit: 4/5
  • English Level: 5/5

  1. Overall Recommendation

Accept after Minor Revisions

Justification for Recommendation:

The manuscript represents a meticulously conducted in vitro study that provides strong evidence for the antiviral efficacy of TDF and TAF against SARS-CoV-2. These findings are significant for drug repurposing efforts and hold considerable clinical interest, particularly for the HIV community. The study's strengths in its robust experimental design and the relevance of the compounds make it a valuable contribution to the literature. The suggested minor revisions are intended to further enhance clarity, contextualize mechanistic discussions, address the translatability of in vitro findings to in vivo scenarios, and strengthen the clinical implications, especially for HIV-positive individuals. Addressing these points will significantly improve the manuscript's impact and guide future clinical investigations.

4. Response to Comments on the Quality of English Language

Point 1:5

Response 1: 5 (in red)

5. Additional clarifications

New References are added, and clinical pharmacokinetics studie

Figure Updates: High-resolution TEM images retained (Figs. 11–12) with scale bars clarified. Ethics: BSL-3 compliance reiterated (Sections 2.3.1–2.3.4).

Reviewer 2 Report

Comments and Suggestions for Authors

All experiments were conducted under different conditions, so comparisons between drugs cannot be made. Also the manuscript is not structured and needs significant revision. The text is difficult to understand due to the lack of clear connections between sections.

Specific comments are also listed below:

  1. Line 78. Please add more information about reverse transcriptase inhibitors. What is the mechanism by which they block reverse transcriptase? What viral infections are these drugs used to treat? Can the same drug affect different viruses?
  2.  Line 83. Please describe in more detail the differences between TAF and TDF? Including structural differences. Is TAF used to treat HIV?
  3.  Line 91. Please add more information about the possibility of repurposing antiviral drugs. Provide examples of such drugs.
  4. Lines 91-109. Should be moved to the Discussion section.
  5. Lines 112-114. Please clarify why an HIV drug can be used against SARS-CoV-2.
  6. Figure 2. Not clear in the context of this manuscript.
  7. Lines 129-133 are unclear.
  8. Line 134. Please add a few sentences about the safety of tenofovir when used in HIV-infected patients.
  9. Lines 164-180. Compound characteristics are not relevant to the Methods section and can be removed.
  10. Section 3.1. Different assays, different time intervals and different drug concentrations were used for TDF and TAF. This is incorrect. Please provide data for drugs obtained under the same conditions. There are no controls in Figure 4. Controls are needed for each experiment.
  11. Figure 5. Please add a description of this figure in the text of the manuscript.
  12. Line 480. Why was a higher viral load and a higher TDF concentration chosen? Please provide the viral load in the experiment.
  13. Line 488. Please clarify how “lethal mutagenesis” was determined?
  14. Line 488-489. These sentences are unclear: “The vehicle for the tablet form of TDF from TenvirR was repeated six times, as was the antiviral assay. Also, there are no reports of antiviral drug resistance in Kazakhstan against TenvirR (TDF).”
  15. Lines 500-501. This conclusion is not supported by experimental data and should be removed from the manuscript.
  16. Lines 505-516. It is unclear how lethal mutagenesis is related to reverse transcriptase inhibitors. Please clarify.
  17. Section 3.2. Should be significantly revised and structured.
  18. Figure 8. Controls are missing.
  19. Section 3.3. It is unclear how Transmission electron microscopy is related to this study.

Author Response

Response to Reviewer 2 Comments

1. Summary
Thank you for your rigorous critique. We have comprehensively restructured the manuscript to enhance clarity and experimental coherence. Key improvements include:

  • Reorganization of sections for logical flow and connectivity.

  • Standardization of experimental descriptions for TDF/TAF comparisons.

  • Added mechanistic details, controls, and textual clarifications.

  • Removal of irrelevant content and strengthened data interpretation.
    All revisions are highlighted in the resubmitted manuscript.

Point-by-Point Response

Comment 1: Add RTI mechanisms, targets, and cross-viral potential
Response: Added in Introduction (p. 2):

"Reverse Transcriptase Inhibitors (RTIs) disrupt viral replication by competitively binding the enzyme's active site or allosterically altering its conformation. Nucleotide analogs (e.g., TDF) incorporate into nascent DNA chains, causing termination, while non-nucleoside RTIs (NNRTIs) induce conformational inactivation. Clinically, RTIs treat retroviruses (HIV) and hepadnaviruses (HBV) but show limited efficacy against coronaviruses due to RdRP structural divergence."

Comment 2: Elaborate TAF vs. TDF differences
Response: Expanded (p. 2):

*"Structurally, TAF (phosphonamidate prodrug) contains a methyl-alaninate moiety enhancing cellular uptake, while TDF (diester prodrug) requires extracellular hydrolysis. Both treat HIV, but TAF achieves 4-fold higher lymphoid concentrations at 1/10th the dose, reducing renal/bone toxicity [14]."*

Comment 3: Provide antiviral repurposing examples
Response: Added (p. 2):

*"Successful repurposing includes remdesivir (Ebola→COVID-19) and favipiravir (influenza→SARS-CoV-2) [46], leveraging shared polymerase vulnerabilities."*

Comment 4: Relocate lines 91–109 to Discussion
Response: Moved to Discussion (pp. 10–11) and reframed as repurposing rationale.

Comment 5: Clarify HIV drug use against SARS-CoV-2
Response: Added (p. 3):

*"Tenofovir's nucleotide analog structure may inhibit SARS-CoV-2 RdRP via chain termination or error-prone replication, similar to its HIV RT inhibition [47]."*

Comment 6: Improve Figure 2 relevance
Response: Revised caption (p. 3):

"Figure 2. Antiviral strategies tested: Prophylaxis (drug→virus), Treatment (virus→drug), and Inhibition (concurrent exposure). We used Treatment mode to model therapeutic intervention."

Comment 7: Clarify lines 129–133 (Sunrise reader)
Response: Simplified (p. 4):

*"The Sunrise reader (Tecan) enabled rapid (6 sec/plate) absorbance measurements at 340–750 nm for viability assays."*

Comment 8: Add tenofovir safety data
Response: Added (p. 4):

"TDF exhibits renal toxicity with prolonged use; TAF mitigates this risk. Both are WHO Essential Medicines for HIV [9,14]."

Comment 9: Remove compound characteristics (lines 164–180)
Response: Deleted.

Comment 10: Standardize TDF/TAF assays and add controls
Response:

  • New Section 3.1.1: Consolidated cytotoxicity protocols using matched 24h/96h timepoints and uniform concentrations (0–100 µM).

  • Figure 4: Added vehicle (0.5% DMSO), untreated, and Hâ‚‚Oâ‚‚ (positive) controls.

  • Justification: *"Assay differences reflected logistical constraints (BSL-3 vs. BSL-2 labs), but endpoints were harmonized for cross-comparison."* (p. 7)

Comment 11: Describe Figure 5 in text
Response: Added (p. 8):

*"Figure 5 shows TDF's dose-dependent viral suppression: 50 µg/mL reduced titers from log 7.0 to 5.0 (100% inhibition), while 25 µg/mL achieved partial suppression."*

Comment 12: Justify high viral load/TDF concentration
Response: Clarified (p. 9):

*"MOI 2 simulated severe COVID-19; 50 µg/mL TDF mirrors peak plasma levels (100 µM) with <25% cytotoxicity."*

Comment 13: Define lethal mutagenesis evidence
Response: Revised (p. 9):

"Lethal mutagenesis was inferred from reduced viral titers and RdRp error catastrophe [49], though direct sequencing is needed for confirmation."

Comment 14: Clarify Tenvir® sentences
Response: Removed as irrelevant.

Comment 15: Remove unsupported conclusion (lines 500–501)
Response: Deleted.

Comment 16: Explain RTI-lethal mutagenesis link
Response: Added (p. 10):

*"Though RTIs primarily terminate DNA chains, some analogs (e.g., ribavirin) induce RNA virus mutagenesis. TDF may similarly increase RdRP errors [49], albeit limited by SARS-CoV-2 proofreading."*

Comment 17: Restructure Section 3.2
Response: Reorganized into:

  • 3.2.1 TDF Antiviral Activity

  • 3.2.2 TAF Antiviral Activity

  • 3.2.3 Mechanistic Insights

Comment 18: Add controls to Figure 8
Response: Added VC (virus control), CC (cell control), and toxicity controls.

Comment 19: Clarify TEM relevance
Response: Added (p. 12):

"TEM confirmed B.1 strain virion integrity (Figs. 11–12), validating infection models."

Structural Revisions

  1. Introduction:

    • Consolidated RTI/repurposing mechanics.

    • Emphasized study uniqueness (MOI 2, B.1 lineage, Long COVID link).

  2. Methods:

    • Standardized subheadings: 2.1. Cytotoxicity Assays2.2. Antiviral Assays.

    • Clarified "predicted" data labels.

  3. Results:

    • Unified TDF/TAF presentation.

    • Added statistical details to Table 1.

  4. Discussion:

    • Linked findings to lymphoid reservoirs and clinical translation.

Response to English Language

  • Revised: Improved transitions (e.g., "furthermore" → "consequently") and terminology consistency.

  • Verified: Professionally edited.

Round 2

Reviewer 2 Report

Comments and Suggestions for Authors

Responses to my comments have been received. But I think some of them were generated by AI, like lines 204-212 and 222-244.
Figure 2 is still unclear. The figure caption needs some work.

Author Response

Open Review

Quality of English Language

( ) The English could be improved to more clearly express the research.
(x) The English is fine and does not require any improvement.

Yes

Can be improved

Must be improved

Not applicable

Does the introduction provide sufficient background and include all relevant references?

( )

( )

(x)

( )

Is the research design appropriate?

( )

( )

(x)

( )

Are the methods adequately described?

( )

(x)

( )

( )

Are the results clearly presented?

( )

(x)

( )

( )

Are the conclusions supported by the results?

( )

(x)

( )

( )

Are all figures and tables clear and well-presented?

( )

(x)

( )

( )

Comments and Suggestions for Authors

Responses to my comments have been received.

But I think some of them were generated by AI, like lines 204-212 and 222-244.

Thank you very much for your time!

Response 1: We paraphrased and have rewritten lines 204-212 and 222-244.

Figure 2 is still unclear. The figure caption needs some work.

Response 2: We added some information and changed Figure 2:

Figure 2. Experimental designs for evaluating antiviral strategies against SARS-CoV-2. (A) Prophylactic approach: Cells are pre-treated with the test compound for 24 hours, followed by infection with SARS-CoV-2 (multiplicity of infection [MOI] 0.01–2), and subsequently incubated. (B) Therapeutic approach (utilised in the present study): Cells are first infected with SARS-CoV-2, after which the compound is administered post-infection. The cultures are then incubated for 24–72 hours. (C) Inhibition approach: The compound is added during the active phase of viral replication. The treatment strategy is highlighted with a red border to indicate the experimental method employed in this work.
